# Processing of the ribosomal ubiquitin-like fusion protein FUBI-eS30/FAU is required for 40S maturation and depends on USP36

Jasmin van den Heuvel[1,2], Caroline Ashiono[1], Ludovic C Gillet[1†], Kerstin Dörner[1,2], Emanuel Wyler[1‡], Ivo Zemp[1], Ulrike Kutay[1]*

[1]Institute of Biochemistry, Department of Biology, ETH Zurich, Zurich, Switzerland; [2]Molecular Life Sciences Ph.D. Program, Zurich, Switzerland

*For correspondence: ulrike.kutay@bc.biol.ethz.ch

Present address: †Institute of Molecular Systems Biology, Department of Biology, ETH Zurich, Zurich, Switzerland; ‡Max-Delbrück-Center for Molecular Medicine in the Helmholtz Association (MDC), Institute for Medical Systems Biology (BIMSB), Berlin, Germany

Competing interests: The authors declare that no competing interests exist.

**Abstract** In humans and other holozoan organisms, the ribosomal protein eS30 is synthesized as a fusion protein with the ubiquitin-like protein FUBI. However, FUBI is not part of the mature 40S ribosomal subunit and cleaved off by an as-of-yet unidentified protease. How FUBI-eS30 processing is coordinated with 40S subunit maturation is unknown. To study the mechanism and importance of FUBI-eS30 processing, we expressed non-cleavable mutants in human cells, which affected late steps of cytoplasmic 40S maturation, including the maturation of 18S rRNA and recycling of late-acting ribosome biogenesis factors. Differential affinity purification of wild-type and non-cleavable FUBI-eS30 mutants identified the deubiquitinase USP36 as a candidate FUBI-eS30 processing enzyme. Depletion of USP36 by RNAi or CRISPRi indeed impaired FUBI-eS30 processing and moreover, purified USP36 cut FUBI-eS30 in vitro. Together, these data demonstrate the functional importance of FUBI-eS30 cleavage and identify USP36 as a novel protease involved in this process.

## Introduction

In the majority of eukaryotes, one ribosomal protein (RP) of each subunit is encoded as a linear fusion with an N-terminal ubiquitin (Ub). In addition to genome-encoded polyubiquitin chains, these Ub-RP fusion proteins serve as source of cellular ubiquitin, which is liberated from these precursors by endoproteolytic processing (*Martín-Villanueva et al., 2021*). The released ubiquitin constitutes a major fraction of the ubiquitin pool in both yeast and mammalian cells (*Bianchi et al., 2015*; *Finley et al., 1987*; *Ozkaynak et al., 1987*). From the perspective of the ubiquitin proteasome system, it is not entirely clear why ubiquitin is encoded as a fusion with RPs; but such a configuration seems suited to jointly determine the capacities of the protein degradation and protein synthesis machineries. From the perspective of the fused RPs, their N-terminal ubiquitin moieties seem beneficial as they have been suggested to serve as 'in cis' chaperones for folding and solubility of the respective RP partners, as shown by studies in yeast (*Finley et al., 1989*; *Lacombe et al., 2009*).

In humans, the ribosomal proteins eS31 (RPS27A) and eL40 (RPL40) of the small and large ribosomal subunit, respectively, are synthesized as Ub-RP precursor proteins (*Baker and Board, 1991*; *Kirschner and Stratakis, 2000*; *Lund et al., 1985*), similar to their yeast homologs (*Fernández-Pevida et al., 2016*; *Finley et al., 1989*; *Lacombe et al., 2009*; *Martín-Villanueva et al., 2020*). How cleavage of their ubiquitin portions is linked to the incorporation of the respective RPs into nascent ribosomal subunits is not understood, but it was shown to be important for ribosome functionality in yeast (*Fernández-Pevida et al., 2016*; *Lacombe et al., 2009*; *Martín-Villanueva et al., 2020*). In general, ubiquitin deconjugation is performed by a large family of deubiquitinases (DUBs) (*Clague et al., 2019*; *Eletr and Wilkinson, 2014*; *Komander et al., 2009*). About 100 DUBs exist in

mammalian cells, subdivided into seven distinct families based on their structural organization (*Clague et al., 2019*; *Komander et al., 2009*). Biochemical experiments have identified several candidate DUBs for processing of the two human Ub-RP fusions, including UCHL3, USP9X, USP7, USP5 and OTULIN, which may act post-translationally in a partially redundant fashion (*Grou et al., 2015*).

Interestingly, humans and other holozoan organisms, a group not containing yeasts, synthesize a third RP as a fusion protein, namely eS30 (RPS30/FAU). Yet, in case of eS30, its N-terminal fusion partner is not ubiquitin but the ubiquitin-like protein (UBL) FUBI (also known as MNSFβ, FUB1, or UBIM). The coding sequence of FUBI-eS30 was originally identified in the transforming retrovirus FBR-MuSV (Finkel-Biskis-Reilly Murine Sarcoma Virus), which coined the term *Fau* (*F*BR-MuSV-*a*ssociated *u*biquitously expressed) for the FUBI-eS30 coding gene in mammals (*Kas et al., 1992*). FUBI displays 36% identity with ubiquitin. Similar to ubiquitin, FUBI possesses a C-terminal diglycine motif, which is crucial in ubiquitin for its conjugation to and removal from substrates including ubiquitin itself (*Drag et al., 2008*; *Sloper-Mould et al., 2001*). However, in contrast to ubiquitin, FUBI contains only a single internal lysine residue, corresponding to the least accessible lysine residue 27 of ubiquitin (*Castañeda et al., 2016*). Consistently, FUBI chain formation has not been reported, and enzymes mediating conjugation of FUBI to other proteins, akin to those that mediate protein ubiquitylation, remain to be identified (*Cappadocia and Lima, 2018*; *van der Veen and Ploegh, 2012*). Some studies suggested a role for FUBI in the immune system of mammals, that is in the regulation of T cells and macrophages. Such functions may entail the conjugation of FUBI to partner proteins such as the pro-apoptotic factor Bcl-G and the endocytosis regulator endophilin A2 (*Nakamura et al., 2015*; *Nakamura and Shimosaki, 2009*; *Nakamura and Yamaguchi, 2006*). However, since FUBI-eS30 is conserved in the holozoan subclade of Opisthokonta, which includes animals and their closest single-celled relatives, although not yeasts (*Aleshin et al., 2007*), the cellular immune system of multicellular organisms is unlikely to be the major determinant for the presence of FUBI.

Of the FUBI-eS30 fusion protein, only eS30 becomes a part of mature ribosomes. It is positioned at the shoulder of the small ribosomal subunit, tucked in between helices h16 and h18 of the 18S rRNA (*Figure 1A*). Three conserved basic residues near the C terminus of eS30 line the mRNA channel, where they may interact with the mRNA phosphate backbone (*Rabl et al., 2011*). The N-terminal end of eS30 extends towards the A-site decoding center that is located near the base of h44. Therefore, a FUBI moiety attached to the N terminus of eS30 may sterically obstruct the access of tRNAs to the ribosome. However, it has remained enigmatic which enzyme separates FUBI and eS30, and how FUBI cleavage is coordinated with the biogenesis of the 40S ribosomal subunit. Also, the exact consequences of a lack of FUBI removal on ribosome function or synthesis are currently unclear.

Here, we set out to address these open questions, exploiting uncleavable FUBI-eS30 constructs expressed in human cells. Persistence of FUBI on nascent 40S subunits impaired late cytoplasmic steps of 40S maturation, leading to the accumulation of subunits that fail to join the pool of translating ribosomes. Differential proteomics of factors associated with wild-type versus cleavage-deficient FUBI-eS30 mutants identified the nucleolar deubiquitinase USP36 as a candidate processing factor, which was proven by depletion and biochemical reconstitution experiments. Collectively, we demonstrate that removal of FUBI from eS30 is important for 40S subunits to gain competence in protein translation and show that USP36 is a promiscuous enzyme whose activity extends from ubiquitin to the ubiquitin-like protein FUBI.

## Results

To study the importance of FUBI-eS30 processing for ribosome biogenesis and function, we set out to explore the consequences of expressing cleavage-deficient FUBI-eS30 mutants. We reasoned that cleavage of the FUBI-eS30 fusion protein by an unidentified ubiquitin-like protease (ULP) could be impaired by mutation of FUBI's C-terminal diglycine motif, akin to analogous mutations in ubiquitin that are known to affect endoproteolytic cleavage of linear ubiquitin fusion proteins (*Bachmair et al., 1986*; *Butt et al., 1988*; *Lacombe et al., 2009*; *Martín-Villanueva et al., 2020*). We generated tagged FUBI-eS30-StHA (tandem Strep II-hemagglutinin tag) wild-type (WT) or diglycine mutant constructs in which either both glycines were changed to alanines (G73,74A; AA) or the C-terminal glycine was changed to valine (G74V; GV) (*Figure 1B*), inspired by mutations previously

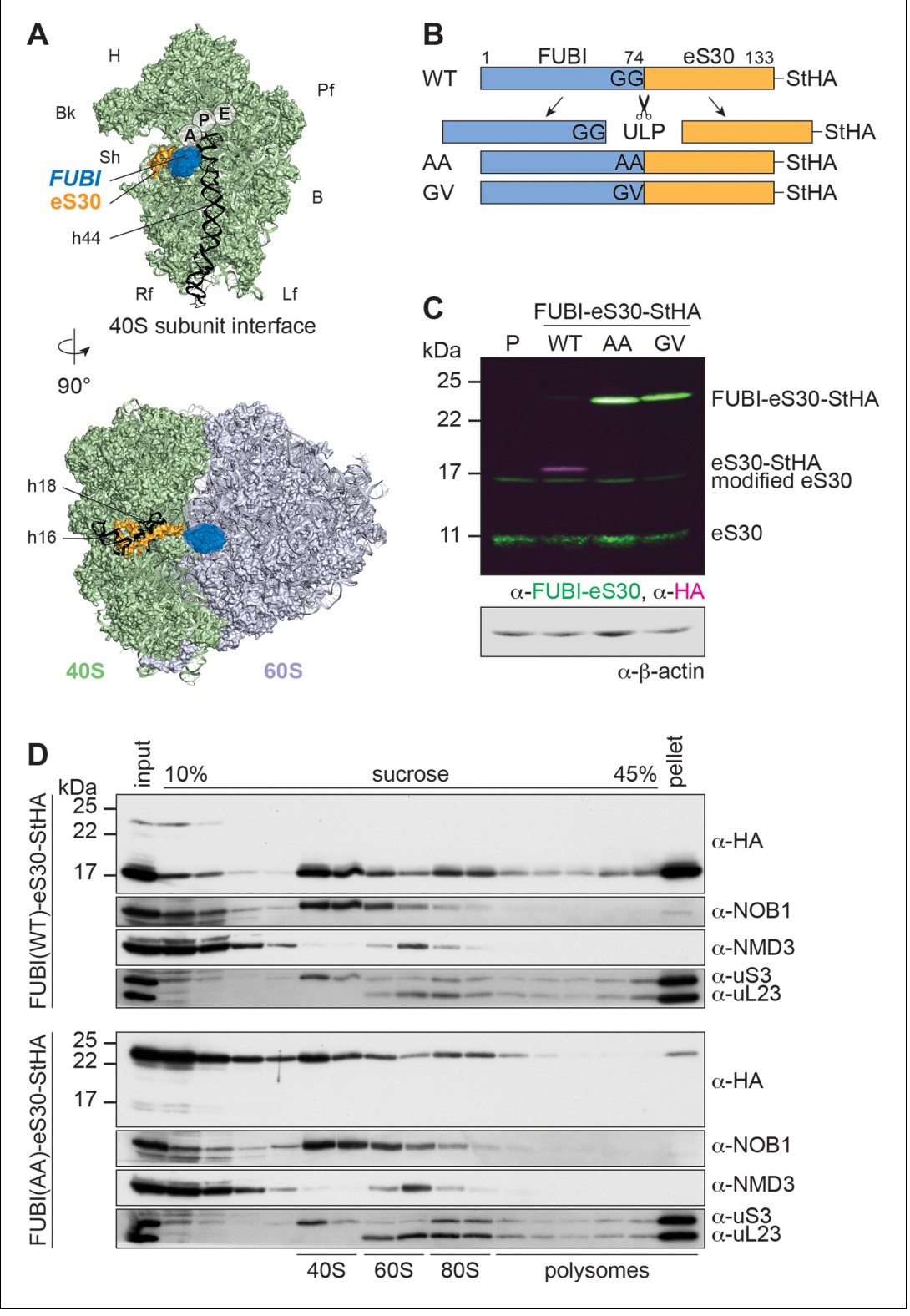

**Figure 1.** Mutant, non-cleavable FUBI-eS30 can be incorporated into (pre-)40S particles. (**A**) Structures (PDB ID: 4UG0 *Khatter et al., 2015*) of the human 40S subunit shown from the subunit interface (top) and of the 80S ribosome viewed from the mRNA entry channel (bottom), highlighting the *hypothetical* location of FUBI (blue) at the N terminus of eS30 (orange). FUBI (PDB ID: 2L7R, state 1 (*Lemak, 2011*), marked with an opaque blue shape) was manually positioned in PyMOL avoiding molecular contacts in the surface representation mode. Ribosomal A-,

*Figure 1 continued on next page*

*Figure 1 continued*

P-, and E-sites are indicated and the 18S rRNA helices h44 (G1702–C1833), h18 (U595–A641), and h16 (C527–G558) are highlighted in black. B, body; Bk, beak; H, head; Lf, left foot; Pf, platform; Rf, right foot; Sh, shoulder. (B) Schematic representation of C-terminally StHA-tagged FUBI-eS30 wild-type (WT) and mutant G73,74A (AA) and G74V (GV) constructs used in this study. The C-terminal diglycine motif of FUBI (G73,G74) was mutated to impair FUBI removal by ubiquitin-like protease(s) (ULP). (C) Immunoblot of tetracycline-inducible HeLa cell lines expressing the indicated FUBI-eS30-StHA constructs using anti-FUBI-eS30/FAU and anti-HA primary antibodies. Fluorescent secondary antibodies against anti-FUBI-eS30/FAU (green) and anti-HA (magenta) antibodies were detected simultaneously. P, parental cells. Note that the StHA tag adds ~7 kDa to the (FUBI-)eS30 protein constructs. eS30 runs at a higher MW than the expected 7 kDa, likely due its high content in positively charged residues. (D) Extracts from FUBI-eS30-StHA WT (top) or AA mutant (bottom) HEK293 cell lines were separated on a linear 10–45% sucrose gradient by centrifugation. Expression of the FUBI-eS30-StHA constructs was induced with 0.1 µg/ml tetracycline for 17 hr. Input, gradient, and pellet fractions were analyzed by immunoblotting using the indicated antibodies.

The online version of this article includes the following source data for figure 1:

**Source data 1.** Source data for *Figure 1C and D* with relevant bands labeled on the uncropped original blots.

introduced into Ub-RP fusion proteins in budding yeast (*Lacombe et al., 2009*). Using these constructs, we produced tetracycline-inducible HeLa and HEK cell lines in a cellular background containing endogenous FUBI-eS30 to uncouple the study of FUBI-eS30 processing from effects that could arise from lack of free FUBI or eS30. While most of FUBI(WT)-eS30-StHA was processed, alterations of the diglycine motif rendered both mutant constructs non-cleavable (*Figure 1C*). This indicates that a FUBI-eS30-specific ULP also requires an intact diglycine motif for processing.

To test whether the non-cleavable mutants can be incorporated into ribosomal subunits and translating ribosomes, we analyzed the sedimentation behavior of the WT and AA mutant constructs in sucrose gradients of lysates prepared from inducible HEK293 cell lines (*Figure 1D*). WT FUBI-eS30-StHA was cleaved and successfully incorporated into small ribosomal subunits that can engage in mRNA translation, evident by its co-sedimentation with the 40S-specific ribosome biogenesis factor (RBF) NOB1 in the 40S peak, and with the RPS uS3 in the 40S, 80S and polysome-containing fractions. In contrast, a larger portion of FUBI(AA)-eS30-StHA was migrating in lighter gradient fractions that contain monomeric proteins and small protein complexes, suggesting that its incorporation into 40S subunits could be less efficient. Still, FUBI(AA)-eS30-StHA was also detected in 40S- and 80S-containing fractions, indicating that the mutant fusion protein can in principle be incorporated into ribosomal subunits. Importantly, however, its levels were decreased in gradient fractions containing actively translating ribosomes and the polysome pellet. Thus, while FUBI removal is not essential for incorporation of eS30 into pre-40S particles, it appears to be a prerequisite for proper function of 40S subunits in translating polysomes.

To investigate the effects of non-cleavable FUBI-eS30-StHA mutant constructs on ribosomal subunit ratio and translation, we next performed polysome profile analysis of HEK293 cells expressing FUBI-eS30 constructs (*Figure 2A*). While there were no significant differences between the $A_{254}$ profiles of cell extracts obtained from parental HEK293 cells and cells expressing FUBI(WT)-eS30-StHA, expression of the non-cleavable FUBI-eS30-StHA AA and GV mutants led to significantly decreased 40S and polysome levels (*Figure 2B*). Concomitantly, free 60S subunit levels were increased, which is commonly observed if eukaryotic 40S production is impaired such as upon depletion of various RPS or 40S RBFs (*Cheng et al., 2019*; *O'Donohue et al., 2010*). Thus, defective maturation of pre-40S subunits containing mutant, non-cleavable FUBI-eS30 could be the reason why this fusion protein does not appear in polysomes (*Figure 1D*).

## Expression of non-cleavable FUBI-eS30 mutants induces pre-18S rRNA processing defects

To better define the nature of the observed 40S ribosome biogenesis defect, we first compared pre-rRNA processing in HeLa cells expressing either WT or mutant FUBI-eS30 constructs by Northern blot analysis (*Figure 2C*). The processing of the initial 47S pre-rRNA occurs in a sequential manner along at least two parallel pathways (*Henras et al., 2015* and *Figure 2C*). To excise the 18S rRNA, the polycistronic 47S precursor is shortened at the 5' and 3' ends and cleaved within the spacer

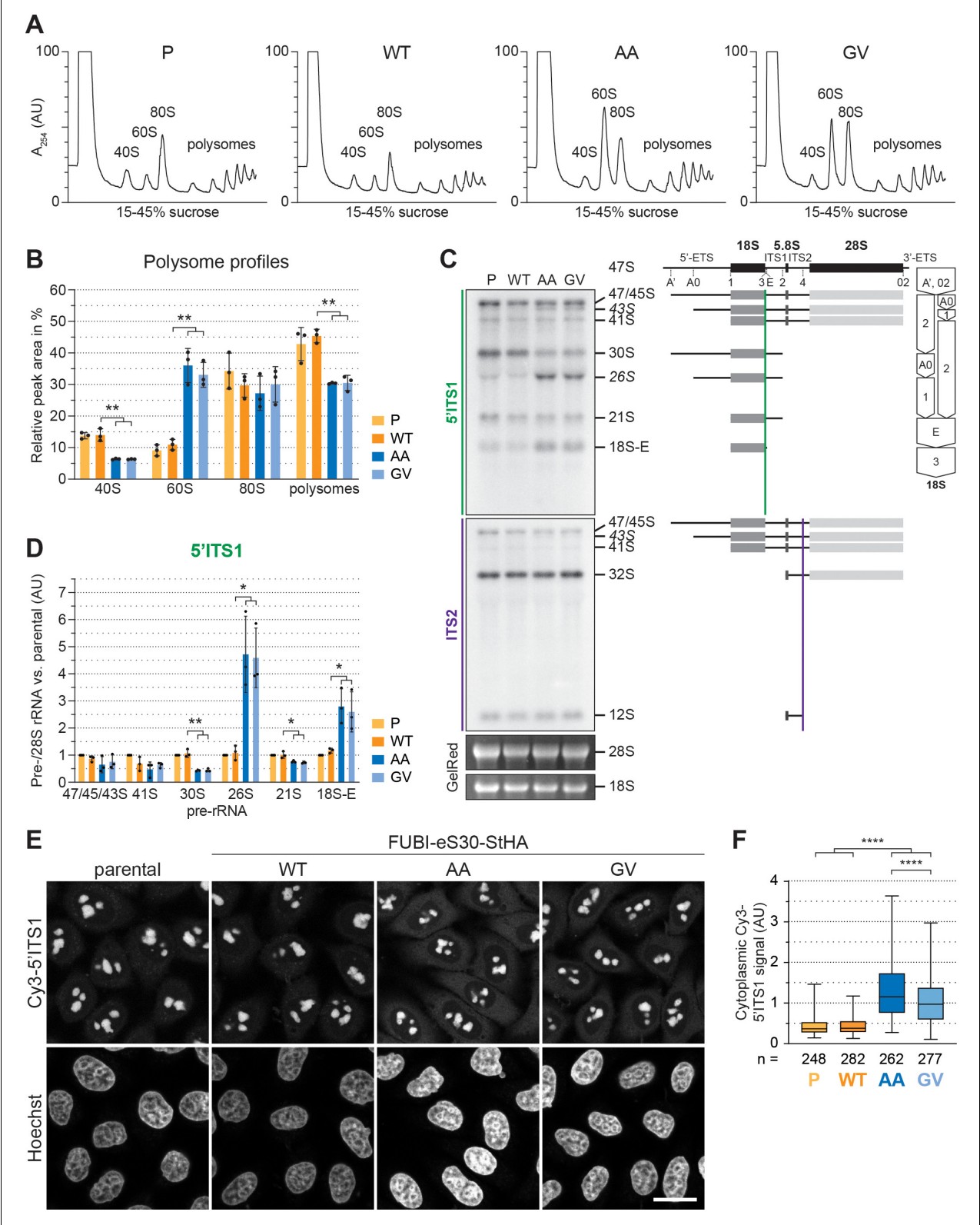

**Figure 2.** 40S ribosome biogenesis is defective upon expression of non-cleavable FUBI-eS30 mutants. (**A**) Extracts from parental (P), wild-type (WT) or mutant (AA, GV) FUBI-eS30-StHA HEK293 cell lines were separated on a linear 15–45% sucrose gradient by centrifugation and analyzed by polysome profiling recording the absorption at 254 nm. (**B**) Quantification of three biological replicates of polysome profiles as shown in (A) determining the relative areas beneath the A₂₅₄ traces of 40S, 60S, 80S, and the first five polysome peaks. Unpaired t-test, mean ± SD, N = 3, **p < 0.01. (**C**) Northern

*Figure 2 continued on next page*

*Figure 2 continued*

blot analysis of total RNA extracted from parental (P), WT or mutant (AA, GV) FUBI-eS30-StHA HeLa cell lines using radioactively labeled probes hybridizing to the 5' region of ITS1 (5'ITS1) or the ITS2. Mature 28S and 18S rRNAs were visualized by GelRed staining of the gel. The short-lived 43S and 26S rRNA precursors are labeled in italics. The rRNA precursors are schematically indicated on the right including the probe binding sites (colored lines) and the processing sites (dashed lines). On the very right, a simplified processing scheme indicates the generation of 18S rRNA precursors by successive endo- and exonucleolytic processing in two parallel pathways according to *Henras et al., 2015*. (D) Quantification of the indicated pre-rRNA species based on the 5'ITS1 signal normalized to mature 28S rRNA in three biological replicates as shown in (C), expressed as fold changes relative to the parental cell line. Unpaired t-test, mean ± SD, N = 3, *p < 0.05, **p < 0.01. Quantification pre-rRNAs based on the ITS2 signal is shown in *Figure 2—figure supplement 1*. (E) Fluorescence in situ hybridization of parental, WT or mutant (AA, GV) FUBI-eS30-StHA HeLa cell lines using a Cy3-labeled 5'ITS1 probe (*Rouquette et al., 2005*). To enhance lower gray levels, γ value was set to 1.5 in parallel for all images. DNA was stained with Hoechst. Scale bar, 20 μm. (F) Quantification of cytoplasmic Cy3-5'ITS1 signals measured in three biological replicates of the experiment shown in (E). Box plots represent the range, quartiles, and mean of the measured signals for the indicated total number of cells (n) per cell line (P, WT, AA, GV). Unpaired t-test, N = 3, n ≥ 46, ****p < 0.0001. Note that, for unknown reason, there is a small but significant difference between the AA and GV mutants.

The online version of this article includes the following source data and figure supplement(s) for figure 2:

**Source data 1.** Source data for *Figure 2C* with relevant areas labeled on the uncropped edited images.

**Source data 2.** Unedited image of the ITS1 blot shown in *Figure 2C*.

**Source data 3.** Unedited image of the ITS2 blot shown in *Figure 2C*.

**Source data 4.** Unedited image of the gel shown in *Figure 2C*.

**Figure supplement 1.** Quantification of pre-rRNAs based on the ITS2 signal.

between the 18S and 5.8S rRNAs. This yields the nucleolar 30S pre-rRNA which is readily detectable by Northern blotting (*Figure 2C*). Additional processing at the 5' end produces the 21S pre-rRNA, which is then further matured at the 3' end via the so-called 18S-E pre-rRNA intermediates in the nucleo- and cytoplasm. We assessed the cellular levels of the various 18S rRNA precursors using a probe hybridizing to the 5' region of the internal transcribed spacer 1 (5'ITS1). Whereas expression of FUBI(WT)-eS30-StHA did not significantly alter the levels of 18S rRNA precursors compared to parental cells, the non-cleavable mutants induced a strong increase in the levels of 26S and 18S-E pre-rRNAs and a significant decrease in the 30S and 21S precursors (*Figure 2D*). This concomitant increase in the levels of both 26S and 18S-E pre-rRNA is reminiscent of their accumulation upon depletion of a subset of RPS termed progression (p-)RPS that are important for 21S and 18S-E processing as well as cleavage at site 3 (*O'Donohue et al., 2010*). In contrast, expression of the non-cleavable mutants did not affect rRNA precursors specific to the 60S subunit (*Figure 2—figure supplement 1*), as we only detected an accumulation of 43S pre-rRNA, a usually short-lived precursor common to both subunits. Thus, the non-cleavable FUBI-eS30-StHA mutants exert dominant-negative effects on 18S pre-rRNA processing, affecting both early and late steps of the processing pathway, with the major defect concerning processing at site 3.

To further characterize these 18S pre-rRNA processing defects, we examined the localization of 18S rRNA precursors by RNA fluorescence in situ hybridization experiments using a Cy3-labeled probe against the 5' end of ITS1 (*Figure 2E*). In parental and WT cells, the 5'ITS1-positive precursors localized to nucleoli (*Rouquette et al., 2005*), where most early processing steps during maturation of pre-40S subunits take place, as expected. 18S-E pre-rRNA-containing 40S particles are then transported from nucleoli to the cytoplasm (*Preti et al., 2013*), where removal of the remaining 3' overhang by NOB1 generates mature 18S rRNA (*Fatica et al., 2003*; *Pertschy et al., 2009*; *Sloan et al., 2013*). Consistent with the increased 18S-E pre-rRNA levels observed by Northern blot analysis, cells expressing the AA or GV mutants showed a significantly increased cytoplasmic Cy3-5'ITS1 signal in addition to the Cy3-positive nucleoli (*Figure 2E and F*), indicating that cytoplasmic processing of 18S-E pre-rRNA was impaired.

## eS30 is incorporated into nuclear pre-40S particles

In general, the timing of pre-40S incorporation of most RPS coincides with the major 40S maturation defects induced by their depletion (*O'Donohue et al., 2010*; *Wild et al., 2010*). However, for depletion of human eS30 (RPS30/FAU), different defects have been reported depending on the applied readout (*O'Donohue et al., 2010*; *Wild et al., 2010*). It has therefore remained difficult to deduce whether eS30 is incorporated into pre-40S subunits in the nucleus or in the cytoplasm. To test

whether eS30 already joins nuclear pre-40S particles, we induced early 40S biogenesis defects by depletion of some key required factors (see below) and checked by immunofluorescence whether StHA-tagged eS30 derived from the FUBI(WT)-eS30-StHA fusion protein accumulates in the nucleus

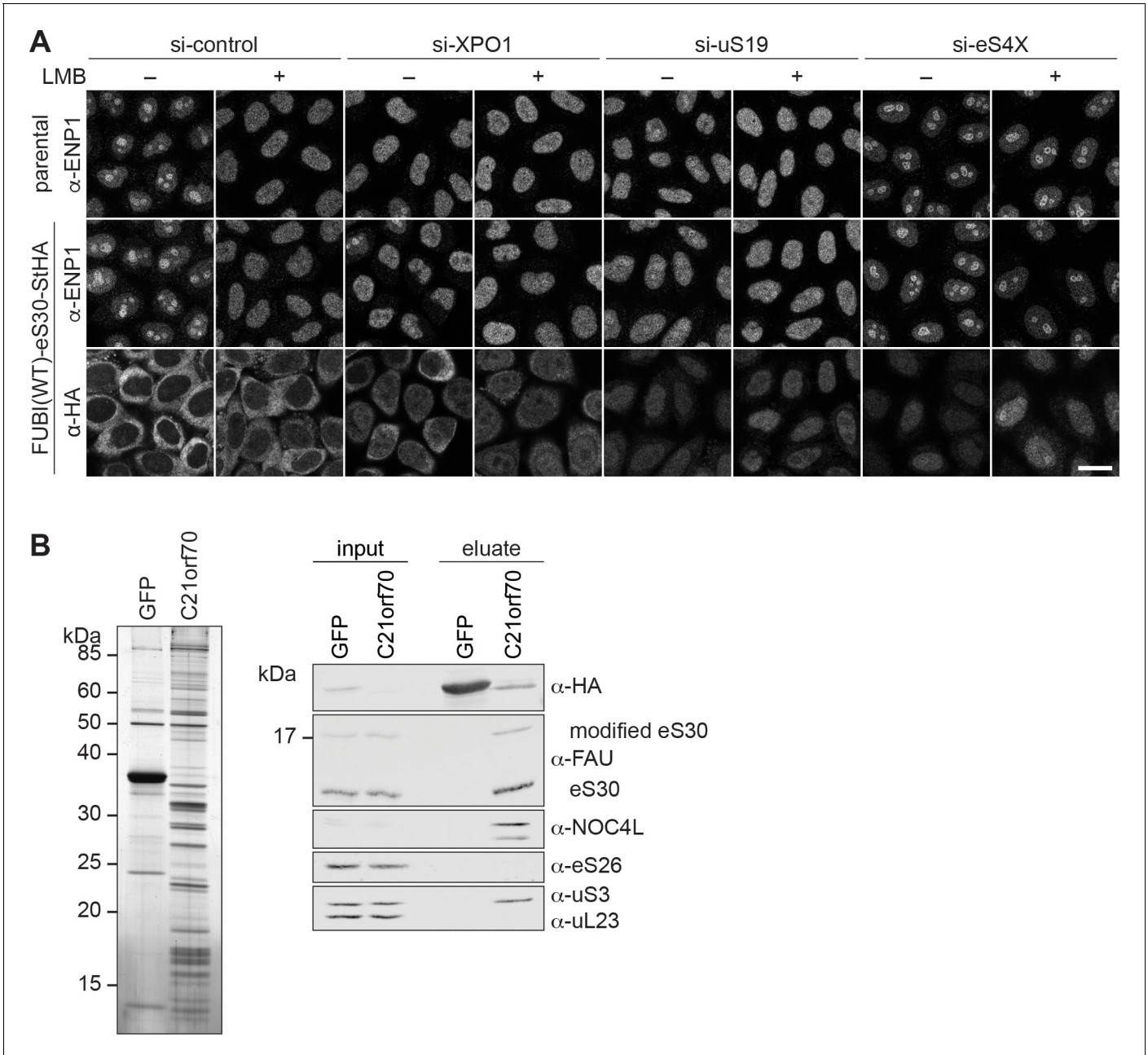

**Figure 3.** eS30 joins pre-40S ribosomal subunits in the nucleus. (**A**) Parental and tetracycline-inducible FUBI(WT)-eS30-StHA HeLa cell lines were treated with the indicated siRNAs (5 nM, 48 hr) and LMB (+; 20 nM, 90 min) before fixation. Cells were immunostained with an antibody against the 40S RBF ENP1. FUBI(WT)-eS30-StHA expressing cells were co-stained with an anti-HA antibody. Scale bar, 20 µm. (**B**) StrepTactin affinity purification of HASt-GFP (GFP) and HASt-C21orf70 (C21orf70) from HEK293 cell lysates. Eluates were analyzed by SDS-PAGE followed by silver staining (left) and together with the input lysates analyzed by immunoblotting using the indicated antibodies (right).

The online version of this article includes the following source data for figure 3:

**Source data 1.** Source data for *Figure 3B* with relevant areas labeled on the uncropped edited image of the gel and with relevant bands labeled on the uncropped original blots.

**Source data 2.** Unedited image of the gel shown in *Figure 3B*.

due to impaired subunit maturation (*Figure 3A*). To control for the efficiency of the respective treatments on 40S subunit maturation, we used the RBF ENP1 (BYSL) as a marker. ENP1 is a shuttling RBF that primarily localizes to nucleoli at steady state (si-control, -LMB) but accompanies maturing 40S subunits to the cytoplasm (*Badertscher et al., 2015*; *Zemp et al., 2009*). Upon treatment of parental HeLa or FUBI(WT)-eS30-StHA expressing cells with the XPO1 (CRM1)-specific nuclear export inhibitor leptomycin B (LMB) that blocks pre-40S export (*Thomas and Kutay, 2003*; *Trotta et al., 2003*), ENP1 accumulated in the nucleoplasm as expected (*Figure 3A*).

In untreated WT cells, eS30-StHA displayed a strong cytoplasmic signal, consistent with its presence in the active pool of ribosomal subunits (*Figure 1D*). Upon LMB treatment, a weak nucleoplasmic eS30-StHA signal became apparent in addition. Similarly, nuclear accumulation of eS30-StHA was also observed upon depletion of the pre-40S export receptor XPO1, suggesting that eS30-StHA is a part of nuclear pre-40S particles that accumulate in the nucleoplasm due to export inhibition. To exclude that this is simply due to the fact that eS30 itself is kept out of the nucleus by XPO1-mediated export, we directly interfered with nuclear 40S biogenesis by depletion of uS19 (RPS15), which is known to be required for late steps of nucleoplasmic pre-40S maturation (*Léger-Silvestre et al., 2004*; *Rouquette et al., 2005*). This led to a prominent nucleoplasmic accumulation of eS30-StHA, indicating that eS30 responds to perturbations of nuclear 40S maturation and is incorporated into nascent subunits in the nucleus. To examine whether eS30 might join pre-40S particles already in nucleoli, we depleted the primary binding RPS eS4X (*Bernstein et al., 2004*; *Lastick, 1980*; *Wild et al., 2010*), which is known to induce early nucleolar pre-40S subunit maturation defects (*Badertscher et al., 2015*; *Wild et al., 2010*). This resulted in the retention of ENP1 in nucleoli in presence of LMB as expected (*Badertscher et al., 2015*). Under these conditions, we observed both a nucleoplasmic and a nucleolar eS30-StHA signal, especially in presence of LMB, which could suggest that eS30 is targeted to nucleoli for its incorporation into pre-40S subunits. Collectively, the nucle(ol)ar accumulation of FUBI(WT)-eS30-StHA upon induction of nuclear 40S maturation and export defects suggests that eS30 assembles into nuclear pre-40S ribosomes, likely already in the nucleolus. This conclusion is supported by previous proteomic data that indicated an efficient accumulation of newly synthesized FUBI-eS30 in isolated nucleoli (*Lam et al., 2007*). To directly test whether eS30 is a component of nuclear pre-40S, we examined its association with pre-40S particles isolated by pull-down of a HASt-tagged version of the nucle(ol)ar RBF C21orf70 (FAM207A) from HEK293 cells (*Zemp et al., 2014*). Analysis of the pull-down eluates by immunoblotting revealed that eS30 is indeed associated with these nucle(ol)ar 40S precursors (*Figure 3B*) that also contained the early pre-40S RBF NOC4L, as expected (*Zemp et al., 2014*), and lacked eS26, which is incorporated into pre-40S subunits in the cytoplasm (*Ameismeier et al., 2020*; *Plassart et al., 2021*).

## Non-cleavable FUBI-eS30 mutants induce defects in late cytoplasmic 40S subunit maturation

We next examined changes in the steady state localization of some key 40S RBFs upon expression of the non-cleavable FUBI-eS30 mutants, as this allows to more clearly pinpoint the step affected by a certain perturbation of 40S maturation. Nucleolar 40S subunit biogenesis gives rise to early pre-40S particles that contain a number of RBFs including ENP1, PNO1 (DIM2), and RRP12, all of which remain associated during transit of premature subunits from nucleoli through the nucleoplasm to the cytoplasm (*Figure 4A*). Further RBFs join in the nucleoplasm before nuclear export, including LTV1, TSR1, the atypical protein kinase RIOK2, and the site 3 endonuclease NOB1. Accordingly, there are various 40S RBFs that accompany maturing subunits from the nucleus to the cytoplasm, which are then successively released in coordination with structural rearrangements that shape the maturing pre-40S particles, supported by additional RBFs such as CK1, RIOK1, and EIF1AD (*Ameismeier et al., 2018*; *Ameismeier et al., 2020*; *Mitterer et al., 2019*; *Plassart et al., 2021*; *Schäfer et al., 2006*; *Widmann et al., 2012*; *Zemp et al., 2014*; *Zemp et al., 2009*).

Immunofluorescence analysis of RIOK2, PNO1, NOB1, and RIOK1, which are among the last factors to be released during the final phase of 40S maturation in the cytoplasm, revealed strong changes in their localization and recycling behavior upon expression of the non-cleavable FUBI-eS30 mutants (*Figure 4B*). Strikingly, PNO1, an RBF that localizes to nucleoli under steady state conditions, relocated to the cytoplasm. Similarly, a prominent recycling defect was also evident for RIOK2 and NOB1. Both are cytoplasmic at steady state but in cells expressing non-cleavable FUBI-eS30-StHA, they failed to accumulate in the nucleus upon LMB treatment, reflecting a failure in their

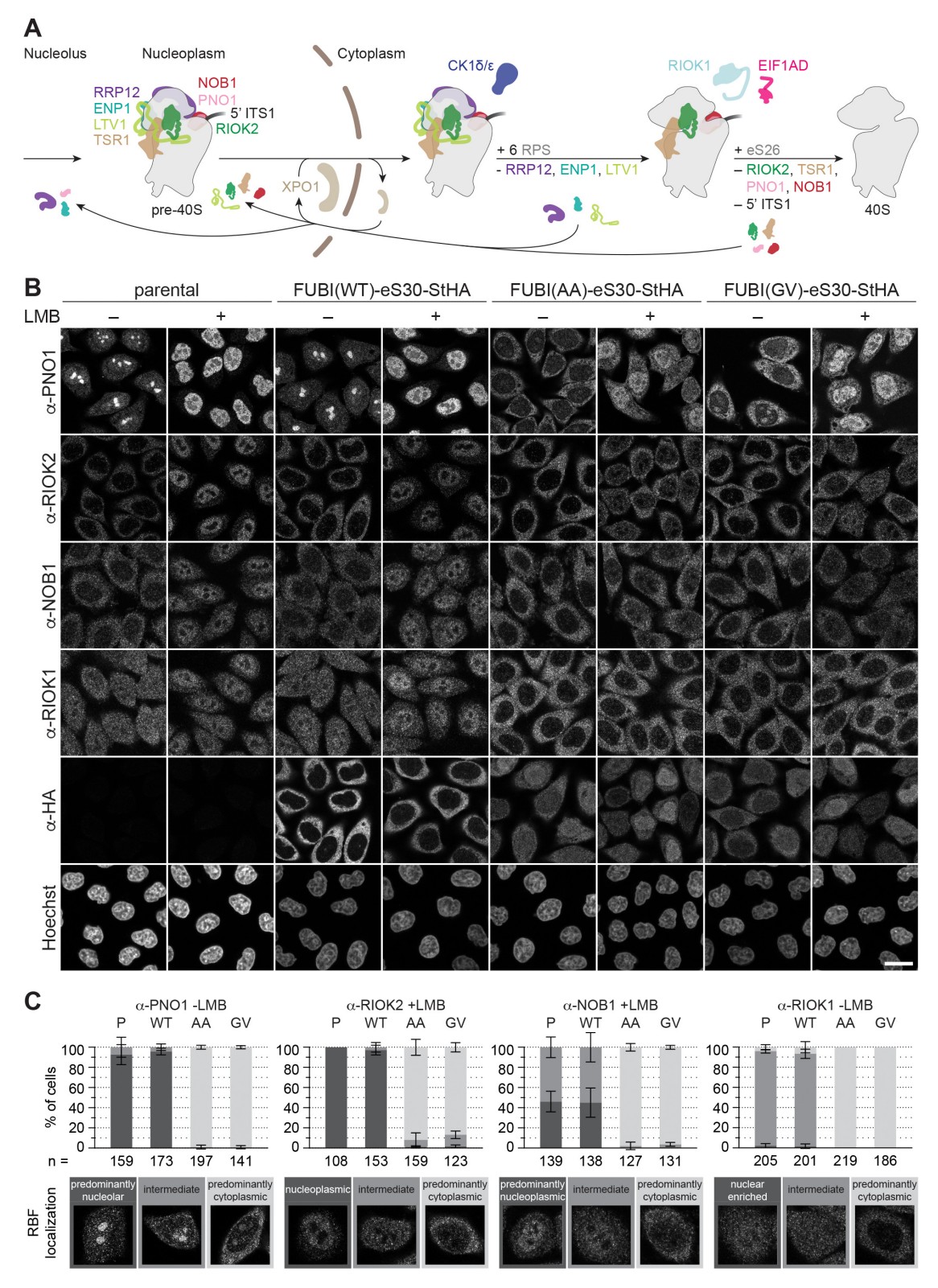

**Figure 4.** Non-cleavable FUBI-eS30 mutants have a dominant-negative effect on late cytoplasmic steps of 40S subunit biogenesis. (**A**) Scheme of pre-40S maturation focusing on late nucleoplasmic to cytoplasmic maturation steps and highlighting the RBFs mentioned in the text. Nucleoplasmic pre-40S particles containing 18S-E pre-rRNA with its characteristic 5' ITS1 remnant and bound to various RBFs can be exported in an XPO1-dependent manner. In the cytoplasm, late-assembling RPS are incorporated, the 3' overhang of 18S rRNA is cleaved off by the endonuclease NOB1, and the RBFs

*Figure 4 continued on next page*

*Figure 4 continued*

are released and recycled. Together, these sequential maturation steps lead to the formation of a mature 40S subunit. (**B**) Immunofluorescence analysis of parental, WT or mutant (AA, GV) FUBI-eS30-StHA HeLa cell lines using the indicated antibodies against the constructs (HA) and the 40S RBFs PNO1, RIOK2, NOB1, and RIOK1. Note that HA and RIOK2 co-immunostaining was performed in parallel with Hoechst staining of DNA. Where indicated, cells were treated with leptomycin B (LMB; 20 nM, 90 min) to inhibit XPO1-mediated nuclear export prior to fixation of cells. Scale bar, 20 μm. (**C**) Quantification of RBF localization for selected conditions of the experiment shown in (B). Percentage of cells assigned to the respective phenotypic classes exemplified below was determined from three or in case of PNO1 -LMB four biological replicates for the indicated total number of cells per condition and cell line (n).

The online version of this article includes the following source data and figure supplement(s) for figure 4:

**Figure supplement 1.** Expression of non-cleavable FUBI-eS30 mutants affects nucleolar but not early cytoplasmic 40S or cytoplasmic 60S biogenesis.

**Figure supplement 1—source data 1.** Source data for *Figure 4—figure supplement 1C* with relevant bands labeled on the uncropped original blots.

**Figure supplement 2.** Non-cleavable FUBI-eS30 mutants induce a persistent 40S ribosomal subunit biogenesis defect.

**Figure supplement 2—source data 1.** Source data for *Figure 4—figure supplement 2C* with relevant bands labeled on the uncropped original blots.

timely release from cytoplasmic pre-40S subunits (*Zemp et al., 2014*; *Zemp et al., 2009*). Also for RIOK1, which localizes to both the nucleus and the cytoplasm, cytoplasmic retention was clearly discernible. The observed changes are indicative for a late, cytoplasmic 40S biogenesis defect in cells expressing the non-cleavable FUBI-eS30 mutants, and consistent with the observed defects in 18S-E pre-rRNA processing (*Figure 2C to F*).

In contrast, those RBFs that are released at an earlier stage of cytoplasmic 40S maturation, namely LTV1, ENP1, and RRP12, remained largely unaffected by the expression of the non-cleavable FUBI-eS30 mutants (*Figure 4—figure supplement 1A*). LTV1, which is cytoplasmic at steady state, accumulated in the nucleoplasm of cells expressing either WT or mutant FUBI-eS30-StHA upon LMB treatment, demonstrating that its recycling is not perturbed by non-cleavable FUBI. For ENP1 and RRP12, two RBFs that predominantly localize to nucleoli and accumulate in the nucleoplasm upon LMB treatment, we observed a slight nucleolar retention in cells expressing the non-cleavable mutants. These changes together with the observed alterations in early pre-rRNA processing (*Figure 2*) could be the result of a so-called 'rebound effect', that is a secondary effect arising from the lack of 40S RBFs like PNO1 in nucleoli due to their strong cytoplasmic retention (*Kofler et al., 2020*; *Zisser et al., 2018*). Lastly, the localization of the 60S RBF NMD3 was not perturbed by mutant FUBI-eS30-StHA expression (*Figure 4—figure supplement 1A*), whereas depletion of the 60S RBF AAMP (Sqt1 in yeast) led to a cytoplasmic retention of NMD3 as expected (*Figure 4—figure supplement 1B and C*).

In summary, the comprehensive examination of the (re)localization of various RBFs suggests that FUBI-eS30 cleavage is a prerequisite for recycling of the RBFs RIOK2, PNO1, NOB1, and RIOK1 in final steps of cytoplasmic 40S maturation. In contrast, recycling of 40S RBFs that are released earlier as well as 60S biogenesis remain by-and-large unperturbed by the presence of FUBI in pre-40S particles.

Interestingly, when analyzing the localization of FUBI-eS30-StHA WT and its mutant derivatives in these experiments (*Figure 4*), we noticed a noteworthy difference: In contrast to the cytoplasmic staining of WT eS30-StHA, both non-cleavable mutants showed a reduced cytoplasmic signal and also localized to the nucleoplasm. This may reflect their hampered incorporation into maturing subunits since we observed a larger pool of non-cleavable FUBI-eS30 mutants in the light fractions of sucrose gradients compared to the mostly cleaved and 40S-associated WT (FUBI-)eS30-StHA (*Figure 1D*). As the partial nuclear accumulation of non-cleavable FUBI-eS30 mutants seemed at odds with the observed 40S maturation defects in the cytoplasm, we analyzed the fate of the nuclear FUBI-eS30 constructs by adding a chase period in tetracycline-free medium (*Figure 4—figure supplement 2A*). After the chase, the nuclear signal vanished and both mutants primarily localized to the cytoplasm like WT FUBI-eS30-StHA. Importantly, however, the late 40S subunit biogenesis defect persisted, as assessed by analysis of PNO1 localization (*Figure 4—figure supplement 2B*). Consistent with these data, immunoblot analysis of polysome profiles revealed that FUBI(AA)-eS30-StHA was mainly particle-associated after the chase period (*Figure 4—figure supplement 2C and E*). The disappearance of FUBI(AA)-eS30-StHA from lighter sucrose fractions could be due to its incorporation into pre-40S particles over time and/or degradation of the unincorporated protein, which may also be reflected by the generally weaker signal in immunofluorescence (*Figure 4—figure*

supplement 2B). Importantly, since PNO1 was still trapped in the cytoplasm of AA- and GV-expressing cells and matching polysome profiles indicated 40S biogenesis defects similar to unchased cells, we conclude that the observed 40S biogenesis defects are primarily caused by pre-40S-associated non-cleavable FUBI-eS30-StHA mutants in the cytoplasm. Although we consider it unlikely, we cannot formally exclude that unincorporated, uncleaved FUBI-eS30 could serve as an unspecific 'sink' for one or some of the affected RBFs. However, at least NOB1 does not change its sedimentation behavior toward the pool of free protein in sucrose gradient analysis (*Figure 4—figure supplement 2C and D*), consistent with our conclusion.

## Identification of FUBI-interacting candidate proteases

To characterize the molecular composition of FUBI-containing particles and to identify candidate FUBI-eS30 protease(s), we performed differential affinity purification mass spectrometry (AP-MS) experiments from lysates of HEK293 cells expressing either WT or non-cleavable FUBI-eS30-StHA constructs (*Figure 5*). We hypothesized that the non-cleavable mutants could serve as enzyme traps and enrich ULP(s), since ULP(s) may be able to bind to but not to cleave the mutant substrates, potentially hampering their release from the cleavage-resistant constructs.

Comparison of the StrepTactin pull-down eluates of the HASt-GFP negative control with the WT and non-cleavable FUBI-eS30-StHA constructs by silver staining confirmed a clean enrichment of ribosomal particles based on the characteristic band patterns (*Figure 5A*, *Montellese et al., 2020*; *Widmann et al., 2012*; *Wyler et al., 2011*). The eluates of three independent experiments were subjected to MS analysis using data-dependent acquisition (*Figure 5B*, *Supplementary file 1*). In all (FUBI-)eS30-StHA pull-downs, ribosomal proteins of both subunits and a variety of 40S RBFs were the most strongly enriched factors (*Supplementary file 1*, *Figure 5A and C*). When inspecting the dataset for candidate FUBI processing enzymes, we found three DUBs to confidently interact with the FUBI-eS30 constructs, namely USP16, USP10, and USP36, all of which belong to the ubiquitin-specific protease (USP) family. While USP16, an established 40S RBF known to remove ubiquitin from eS31 (Rps27a) (*Montellese et al., 2020*), was equally co-purified with both WT and non-cleavable FUBI-eS30-StHA baits, USP10 and USP36 were enriched on the non-cleavable mutants (*Figure 5B*). Importantly, these observations were confirmed by immunoblotting of the pull-down eluates (*Figure 5C*), which made USP10 and USP36 prime ULP candidates for FUBI-eS30 processing.

USP10 is a highly conserved cytoplasmic DUB and has been previously described to interact with 40S subunits (*Clague et al., 2019*; *Kapadia and Gartenhaus, 2019*). It is involved in resolution of stalled translation elongation complexes through removal of the regulatory monoubiquitylations from eS10, uS3, and uS5 (*Garshott et al., 2020*; *Jung et al., 2017*; *Meyer et al., 2020*). In contrast, USP36 is the only known catalytically active DUB exclusively localizing to human nucleoli (*Clague et al., 2019*; *Endo et al., 2009a*; *Scherl et al., 2002*). It has been implicated in the deubiquitylation of various nucleolar proteins, including the catalytic subunit of RNA polymerase I, RPA194 (Rpa190 in yeast; *Peltonen et al., 2014*; *Richardson et al., 2012*), the helicase DHX33 (Dhr2 in yeast; *Fraile et al., 2018*), the 2′-*O*-methyltransferase FBL, and nucleophosmin (NPM/B23, *Endo et al., 2009a*). Interestingly, USP36-mediated deubiquitylation has also been shown to lead to the stabilization of the transcription factor Myc (*Sun et al., 2015*; *Thevenon et al., 2020*), a proto-oncogenic regulator of ribosome and protein synthesis (*van Riggelen et al., 2010*).

Besides the two FUBI-specific candidate proteases, we found several additional proteins enriched in the eluates of pull-downs with the non-cleavable mutants (*Figure 5B*, *Supplementary file 1*). Surprisingly, a number of 60S RBFs (*Fatica and Tollervey, 2002*; *Nissan et al., 2002*; *Woolford and Baserga, 2013*), including NMD3, LSG1, GNL3L, and GNL2, were significantly enriched on the mutants (*Figure 5B and C*), indicating that the pre-40S subunits arrested in their maturation by the presence of FUBI may associate with pre-60S subunits. Immunoblotting indeed confirmed the enrichment of NMD3 and LSG1. Whether this enrichment represents a biologically meaningful interaction, is aberrantly induced by the stalled particles or caused by post-lysis association cannot be deduced from this data.

Among the identified RBFs of the 40S subunit, the protein kinase RIOK1, which is usually harder to capture on pre-40S subunits than other 40S RBFs (*Widmann et al., 2012*), was clearly enriched on the non-cleavable mutants (*Figure 5C*), consistent with our recycling data (*Figure 4*). Interestingly, we also observed a shift in migration of RIOK2 induced by the expression of the non-cleavable FUBI-eS30 constructs, potentially caused by the arrest of an activated, phosphorylated form of the kinase

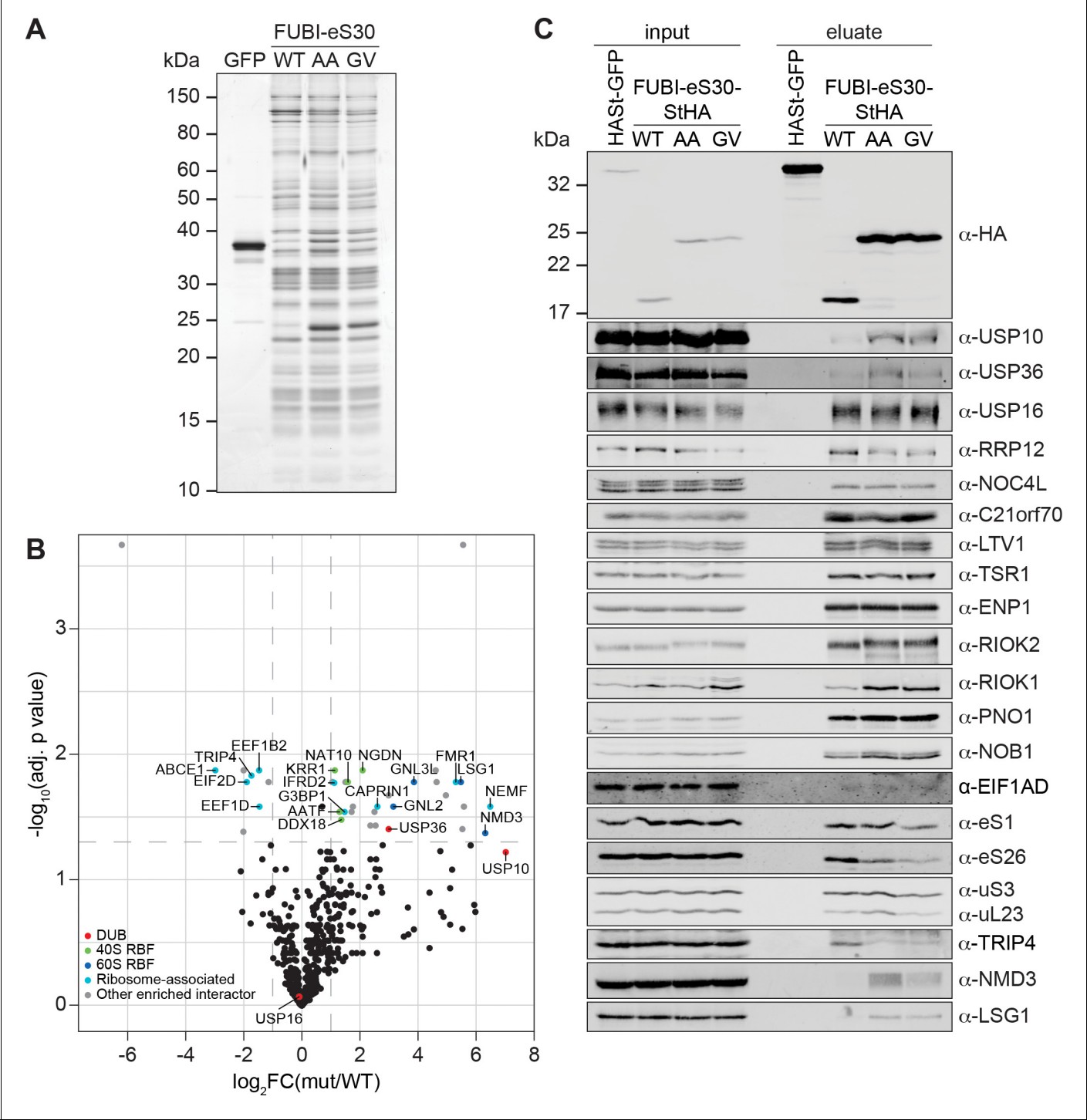

**Figure 5.** Identification of candidate proteases by differential affinity purification mass spectrometry of FUBI-eS30-StHA constructs. (**A**) StrepTactin affinity purification of HASt-GFP (GFP), WT or mutant (AA, GV) FUBI-eS30-StHA from HEK293 cell lysates. Eluates were analyzed by SDS-PAGE followed by silver staining or mass spectrometry. (**B**) Results of the proteomic analysis of three biological replicates as in (A). The spectral counts of proteins that were compared to HASt-GFP confidently enriched on FUBI-eS30-StHA baits (SAINT Bayesian false discovery rate < 1%) were normalized to their size in amino acids. The $\log_2$ fold change of the average number of the spectral counts of the confident interactors identified on the non-cleavable AA and GV mutants (mut) vs. WT FUBI-eS30-StHA ($\log_2$FC(mut/WT)) are plotted against the negative $\log_{10}$ of the adjusted p value (-$\log_{10}$(adj. p value)). All confidently identified DUBs are labeled, significantly enriched interactors (|$\log_2$FC| > 1 and adj. p value < 0.05, demarcated with dashed lines) are categorized as indicated and individual proteins are labeled. Data before and after normalization and filtering are shown in *Supplementary file 1*. (**C**) Inputs and eluates of StrepTactin affinity purification performed as in (A) were analyzed by immunoblotting using the indicated antibodies.

*Figure 5 continued on next page*

*Figure 5 continued*

The online version of this article includes the following source data for figure 5:

**Source data 1.** Source data for *Figure 5A and C* with relevant areas and bands labeled on the uncropped gel and uncropped original blots, respectively.

**Source data 2.** Unedited image of the gel shown in *Figure 5A*.

on pre-40S subunits. Another observation was the striking scarcity of peptides of the recently identi-fied 40S RBF EIF1AD (*Ameismeier et al., 2020*) in the MS data, and it was also barely detectable in the eluates by immunoblotting (*Supplementary file 1*, *Figure 5C*).

## USP36 promotes FUBI-eS30 processing in vivo

We next analyzed whether the identified candidate proteases USP10 or USP36 contribute to FUBI-eS30 processing in living cells. Both DUBs were depleted from HeLa cells by RNAi and cell extracts were analyzed by immunoblotting for accumulation of uncleaved endogenous FUBI-eS30 using an antibody raised against FUBI (*Figure 6A*). Quantification of the level of uncleaved FUBI-eS30 revealed no change upon depletion of USP10 using four different siRNAs. In contrast, the amount of uncleaved FUBI-eS30 was significantly elevated upon treatment of the cells with three of the four USP36-targeting siRNAs. The extent of accumulation of uncleaved FUBI-eS30 varied between the USP36 siRNAs despite similar levels of downregulation. This could either be due to different kinetics of downregulation for the individual siRNAs or to off-target effects altering cellular processes or fac-tors that further attenuate or promote FUBI-eS30 processing.

To ensure that both parts of the FAU fusion protein, that is both FUBI and eS30, jointly accumu-late as the assigned uncleaved protein species in form of an in cis fusion upon USP36 depletion, HEK293 cell lines expressing either N- or C-terminally tagged FUBI-eS30 were treated with control or USP36 siRNA (*Figure 6B*). Upon depletion of USP36, uncleaved tagged FUBI-eS30 accumulated in both cell lines as demonstrated by immunoblotting using an antibody against the HA tag. Collec-tively, these results indicate that separation of FUBI and eS30 depends on USP36 in vivo.

To further strengthen this conclusion by an independent method, we employed a complementary CRISPR interference (CRISPRi) approach (*Boneberg et al., 2019*). We used two different guide RNA sequences to target a gene-silencing, catalytically inactive KRAB-dCas9 protein to the promoter region of USP36 (*Gilbert et al., 2013*; *Horlbeck et al., 2016*; *Qi et al., 2013*). With both guides, USP36 expression was efficiently downregulated, again leading to a significant accumulation of uncleaved endogenous FUBI-eS30 (*Figure 6C*). Importantly, this could be rescued in cell lines expressing EGFP-tagged WT USP36 but not a catalytically inactive mutant, in which the active site cysteine was mutated to an alanine (C131A, CA; *Figure 6C*). In these experiments, we also noted a slight dominant-negative effect of expression of the catalytically inactive USP36 mutant on FUBI-eS30 processing. In conclusion, the CRISPRi-rescue experiment showed that USP36 and its catalytic activity are required for FUBI-eS30 processing.

## USP36 cleaves FUBI fusion proteins in vitro

Since the catalytic activity of USP36 is required for FUBI-eS30 processing in vivo, we finally wished to explore whether USP36 can indeed mediate the cleavage of FUBI-containing fusion proteins in vitro. We expressed both full-length USP36 WT and the CA mutant in insect cells and purified the proteins by StrepTactin affinity purification. Then, we performed in vitro processing assays using purified His$_6$-tagged FUBI(WT)-eS30 or the cleavage-resistant FUBI(AA)-eS30 mutant as substrates sampling different time points during the reaction (*Figure 7A and B*). His$_6$-FUBI(WT)-eS30 was indeed success-fully processed into FUBI and eS30 by wild-type USP36. The cleavage products were already appar-ent at the first time point of 7.5 min and about half of the substrate was turned over after about 20 min at the chosen conditions (*Figure 7A and E*). In contrast, the inactive USP36 CA mutant did not promote processing, excluding that the observed FUBI(WT)-eS30 cleavage was due to any co-purify-ing activity in the enzyme preparation. As expected, the non-cleavable FUBI(AA)-eS30 mutant was resistant to attack by WT USP36 (*Figure 7B*). Together, this data provides direct evidence that USP36 can act as a ULP for FUBI-eS30.

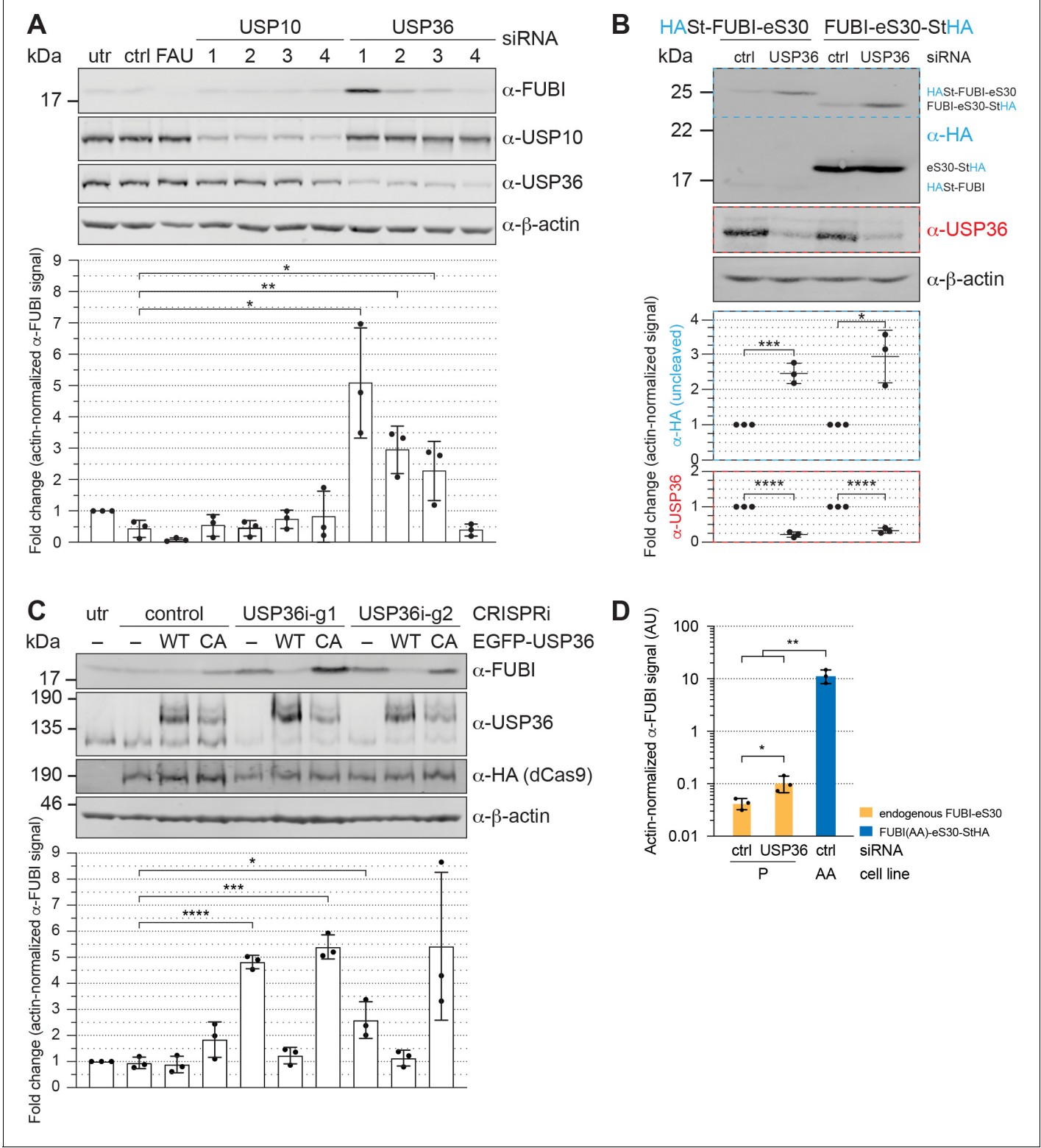

**Figure 6.** USP36 is required for FUBI-eS30 processing in vivo. (**A**) Immunoblot analysis of HeLa K cells that were either left untreated (utr), treated with control (ctrl) siRNA or siRNAs against the indicated factors (48 hr, 10 nM, except for FAU/FUBI-eS30: 5 nM) using the indicated antibodies. Quantification of the fold change of the actin-normalized anti-FUBI signal from three biological replicates shows a significant accumulation of uncleaved FUBI-eS30 upon depletion of USP36 by three different siRNAs. Unpaired t-test, mean ± SD, N = 3, *p < 0.05, **p < 0.01. (**B**) Immunoblot analysis of HASt-FUBI-eS30 and FUBI-eS30-StHA HEK293 cell lines treated with si-control (ctrl) or si-USP36-1 (USP) (20 nM, 48 hr) using the indicated antibodies.

*Figure 6 continued on next page*

*Figure 6 continued*

Note that the linkers between FUBI-eS30 and the tags differ slightly between the two constructs, causing their different running behavior. Actin-normalized anti-HA and anti-USP36 signals in dashed colored boxes of three biological replicates were quantified. Unpaired t-test, mean ± SD, N = 3, *p < 0.05, ***p < 0.001, ****p < 0.0001. (C) Immunoblot analysis of a USP36 CRISPRi-rescue experiment using the indicated antibodies. Parental (–) and EGFP-USP36 WT or catalytically inactive C131A mutant (CA) HeLa cell lines were left untreated (utr) or transfected for 72 hr with a plasmid encoding HA-tagged KRAB-dCas9 and a control single guide RNA (sgRNA) or a sgRNA targeting a site in the promoter region of USP36 (USP36i-g1, USP36i-g2). Expression of the rescue constructs in the respective cell lines was induced 24 hr prior to CRISPRi transfection by addition of tetracycline. Quantification of the fold change of the actin-normalized anti-FUBI signal from three biological replicates confirmed rescue of USP36 depletion by CRISRPi with WT EGFP-USP36 but not the catalytically inactive CA mutant. Unpaired t-test, mean ± SD, N = 3, *p < 0.05, ***p < 0.001, ****p < 0.0001. (D) Parental (P) and FUBI(AA)-eS30-StHA (AA) HeLa cell lines were treated with the indicated siRNAs (si-control (ctrl), si-USP36-1 (USP36); 15 nM, 48 hr). Expression of FUBI(AA)-eS30-StHA was induced with 0.1 μg/ml tetracycline for 24 hr. FUBI immunoblot signals of endogenous, uncleaved FUBI-eS30 (in P) and of FUBI(AA)-eS30-StHA (in AA) were quantified and normalized to actin. Unpaired t-test, mean ± SD, N = 3, *p < 0.05, **p < 0.01.

The online version of this article includes the following source data for figure 6:

**Source data 1.** Source data for *Figure 6* with relevant areas labeled on the uncropped original blots.

USP36 has been classified as a DUB and proposed to deubiquitylate several nucle(ol)ar proteins, thereby preventing their proteasomal degradation (*Endo et al., 2009a*; *Fraile et al., 2018*; *Sun et al., 2015*). With respect to its capacity to process linear ubiquitin fusions, a USP36 fragment encompassing its catalytic domain has been reported to cleave di-ubiquitin substrates in vitro, albeit inefficiently (*Ritorto et al., 2014*). We therefore directly compared the processing activity of USP36 toward FUBI-EGFP and Ub-EGFP fusion proteins (*Figure 7C and D*). Interestingly, both FUBI- and Ub-EGFP were cleaved by USP36 WT but not the CA mutant. Cleavage of FUBI- and Ub-EGFP occurred with by-and-large comparable kinetics (*Figure 7E*), indicating that FUBI is not a preferred substrate in the context of the EGFP fusion proteins. In conclusion, the DUB USP36 is a promiscuous enzyme that can process both linear FUBI and ubiquitin fusion proteins in vitro.

## Discussion

In this work, we have investigated how processing of the exceptional ribosomal protein eS30 that is synthesized as a fusion with the ubiquitin-like protein FUBI, is coordinated with the biogenesis of 40S ribosomal subunits. Our data indicate that eS30 joins pre-40S subunits in the nucleus, where FUBI is cut off from eS30 by the DUB USP36. Non-cleavable mutants of FUBI-eS30 can be incorporated into pre-40S particles, albeit less efficiently than WT, and exert a dominant-negative effect on pre-rRNA processing and recycling of RBFs in the cytoplasm. 40S subunits carrying a persistent FUBI moiety are impaired in entering the pool of translating ribosomes, demonstrating that FUBI processing is an important aspect of 40S subunit maturation.

### FUBI-eS30 cleavage is required for 40S subunit maturation

The expression of non-cleavable FUBI-eS30 mutants causes defects in the cytoplasmic maturation of pre-40S particles, evident by a defect in recycling of a distinctive set of RBFs. The molecular identity of both the affected and unaffected 40S RBFs allows some conclusions on the step(s) of 40S subunit maturation that are perturbed by a failure in FUBI cleavage. Among the studied RBFs, RRP12, ENP1, and LTV1 were normally released from pre-40S particles in the cytoplasm in presence of the non-cleavable FUBI-eS30 mutants. Release of these RBFs is coupled to the maturation of the 40S head, accompanied by the stable incorporation of the uS3–uS10–eS10–uS14 cluster (*Ameismeier et al., 2018*; *Heuer et al., 2017*; *Larburu et al., 2016*; *Mitterer et al., 2019*; *Scaiola et al., 2018*). The lack of recycling defects of this group of RBFs places the impact of FUBI on 40S biogenesis at a downstream step.

In contrast, we observed recycling defects for RIOK2, PNO1, NOB1, and RIOK1. Their stepwise dissociation from cytoplasmic pre-40S subunits accompanies the maturation of the 40S decoding center and the final 3′ end processing of 18S rRNA (*Ameismeier et al., 2020*; *Plassart et al., 2021*), indicating that FUBI impairs these last steps of 40S subunit biogenesis. On the pre-40S subunit, based on the position of the N terminus of eS30, FUBI is expected to occupy a region close to the binding sites of RIOK2, RIOK1, and TSR1 (*Ameismeier et al., 2018*; *Ameismeier et al., 2020*; *Larburu et al., 2016*; *Plassart et al., 2021*), yet none of these factors seemed affected in subunit

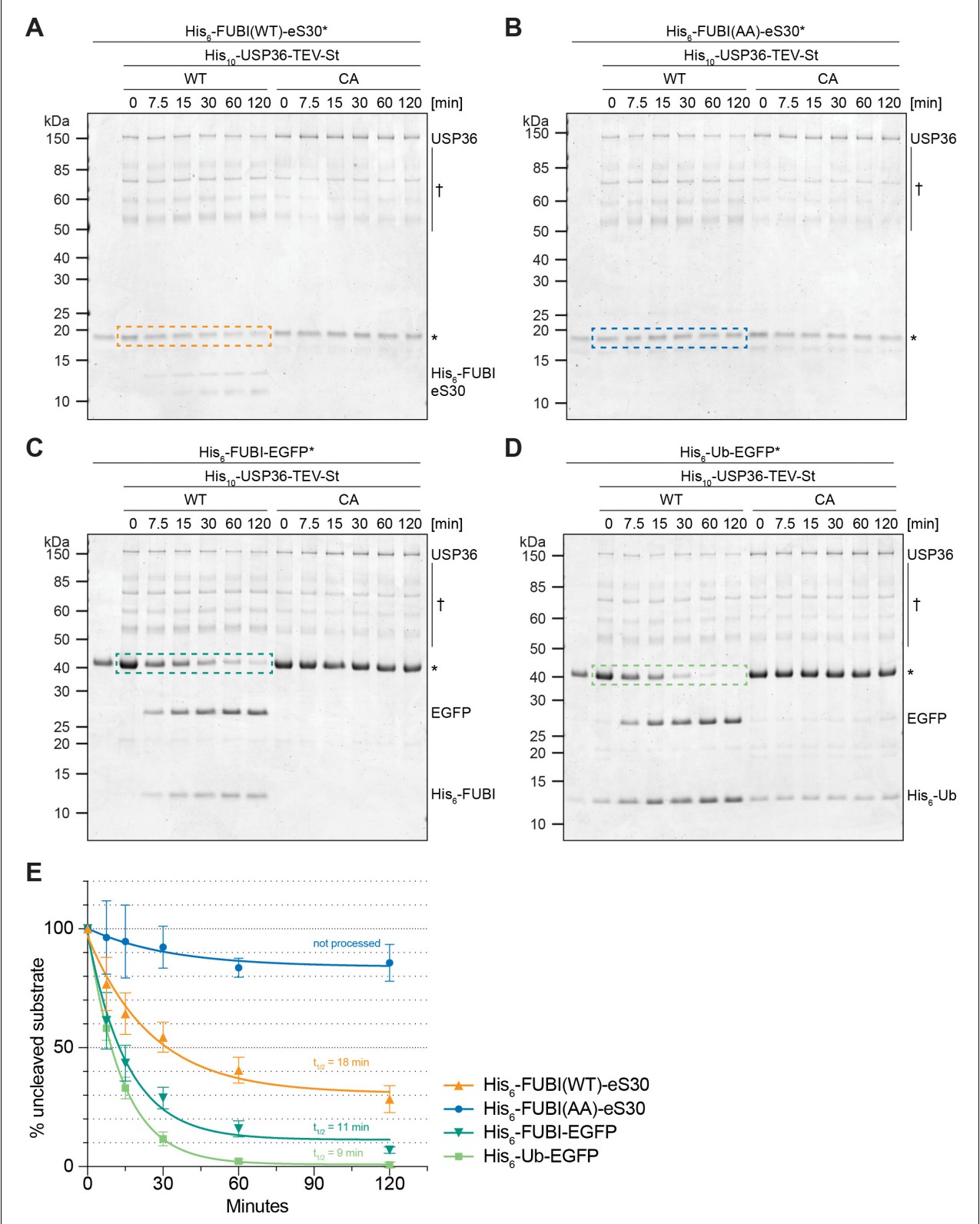

**Figure 7.** USP36 cleaves linear authentic and artificial UB(L) substrates in vitro. In vitro processing assays for which 2.5 μM (**A**) His$_6$-FUBI(WT)-eS30, (**B**) His$_6$-FUBI(AA)-eS30, (**C**) His$_6$-FUBI-EGFP, and (**D**) His$_6$-Ub-EGFP were incubated with 0.5 μM His$_{10}$-USP36-TEV-St WT or CA mutant at 37°C. Samples taken at the indicated time points (0, 7.5, 15, 30, 60, 120 min) were analyzed on Coomassie brilliant blue-stained gels. Unprocessed substrates are marked with an asterisk (*). Note that the enzyme preparation contains USP36 degradation products (marked with a dagger (†), see *Figure 7—figure*

*Figure 7 continued on next page*

*Figure 7 continued*

*supplement 1*). (**E**) Quantification of USP36-dependent processing based on the levels of the uncleaved substrates, highlighted by dashed colored boxes in panels (A to D), each normalized to t = 0 min from three technical replicates. Half-lives ($t_{1/2}$) of fitted one-phase exponential decay curves are indicated for processed substrates.

The online version of this article includes the following source data and figure supplement(s) for figure 7:

**Source data 1.** Source data for *Figure 7* with relevant areas labeled on the uncropped images of the gels.

**Source data 2.** Unedited image of the gels shown in *Figure 7*.

**Figure supplement 1.** Analysis of purified His$_{10}$-USP36-TEV-St.

**Figure supplement 1—source data 1.** Source data for *Figure 7—figure supplement 1* with relevant areas labeled on the uncropped gel and uncropped original blots.

**Figure supplement 1—source data 2.** Unedited image of the gel shown in *Figure 7—figure supplement 1*.

binding based on MS analysis and immunoblotting (*Figure 5*, *Supplementary file 1*). Thus, FUBI must influence 40S maturation by other means. Interestingly, EIF1AD, a recently discovered 40S RBF (*Ameismeier et al., 2020*; *Montellese et al., 2020*), failed to enrich on FUBI-arrested subunits, possibly because FUBI is positioned at the binding site of EIF1AD (*Ameismeier et al., 2020*). It is thus conceivable that impaired recruitment of EIF1AD causes some of the observed recycling defects.

Curiously, although both RIOK2 and RIOK1 are assumed to bind pre-40S particles in a mutually exclusive, successive manner (*Ameismeier et al., 2020*; *Ferreira-Cerca et al., 2014*; *Widmann et al., 2012*; *Zemp et al., 2009*), we observed that the recycling of both kinases was compromised by FUBI. This could indicate that a failure in FUBI processing does not result in the accumulation of one specific precursor species but rather a suite of successive pre-40S particles that are kinetically delayed in their maturation. Alternatively, it cannot be excluded that a single type of particle, on which both RIO kinases are simultaneously present, is trapped in presence of FUBI. Although such a pre-40S state has never been reported (*Heuer et al., 2017*; *Johnson et al., 2017*; *Larburu et al., 2016*; *Strunk et al., 2012*), this assumption would be consistent with the recent observation that incoming RIOK1 binds at the pre-40S shoulder before it eventually flips over to occupy the former position of RIOK2 (*Ameismeier et al., 2020*; *Turowski et al., 2014*). Perhaps, eS30-linked FUBI impedes structural rearrangements of pre-40S subunits at the stage of RIOK2 release, leading to retention of RIOK1 at its landing site. Notably, RIOK2 appears to be shifted to a higher molecular weight in cells expressing the non-cleavable FUBI-eS30 mutants (*Figure 5C*), possibly arrested on pre-40S subunits in an activated phosphorylated state (*Angermayr et al., 2007*; *Geerlings et al., 2003*). This model would eventually explain why depletion of RIOK1 affects RIOK2 dissociation (*Widmann et al., 2012*). Normally, repositioning of RIOK1 is followed by a series of events, including EIF1AD binding, PNO1 release, eS26 incorporation, 18S-E processing, and the subsequent release of NOB1 and RIOK1 itself (*Ameismeier et al., 2020*; *Plassart et al., 2021*), all of which seem affected by uncleaved FUBI. Thus, FUBI removal from eS30 is a crucial event for the final maturation of pre-40S particles.

Surprisingly, we also observed RPLs and an enrichment of several 60S RBFs on non-cleavable FUBI-eS30 mutants. In yeast, the last pre-40S maturation steps have been suggested to involve the formation of 80S-like particles containing pre-40S together with mature 60S ribosomes (*Lebaron et al., 2012*; *Strunk et al., 2012*; *Turowski et al., 2014*). Such particles have not been described for human pre-40S subunits, on which RBFs including RIOK2, RIOK1, potentially TSR1, EIF1AD, and LRRC47 at the subunit interface are expected to counteract premature 60S joining (*Ameismeier et al., 2020*; *Montellese et al., 2020*). While the association of RPLs and 60S RBFs with non-cleavable FUBI-eS30 constructs may point at the formation of 80S-like complexes with (pre-)60S subunits in human cells, it remains unclear whether those present a biochemical artefact or are indeed physiologically meaningful. Clearly, as both subunits are likely immature, the subunit interface in such 80S-like complexes would have to be distinct from that of authentic 80S ribosomes.

## FUBI-eS30 processing is mediated by USP36

Of the 11 known protein-modifying UBLs in humans, FUBI remains the most poorly characterized in terms of (de)conjugating enzymes (*van der Veen and Ploegh, 2012*). To identify candidate FUBI proteases, we used a differential proteomics approach, presuming that ULPs are more readily released from the cleavage products of WT FUBI-eS30 than from the non-cleavable variants. And

indeed, two candidate proteases, USP10 and USP36, were enriched on the non-cleavable mutants (*Figure 5*). Yet, only depletion of USP36 impaired FUBI-eS30 processing, whereas downregulation of USP10 had no effect (*Figure 6A*). Based on these distinct results and the existing literature, it must be assumed that the underlying causes for enrichment of USP10 and USP36 on non-cleavable FUBI-eS30 mutants differ fundamentally.

USP10 is known to remove regulatory monoubiquitylations from eS10, uS3, and uS5 of 40S subunits that are translationally stalled (*Garshott et al., 2020*; *Jung et al., 2017*; *Meyer et al., 2020*). In so doing, it has been suggested to rescue 40S subunits from degradation by ribophagy (*Meyer et al., 2020*). In analogy, one can speculate that the retrieval of USP10 may reflect its binding to cytoplasmic (pre-)40S particles stalled in maturation that are potentially marked by monoubiquitylation due to a failure in FUBI removal.

In contrast, our functional data suggests that USP36 was trapped directly on its processing substrate FUBI-eS30. Several lines of evidence in our data support the conclusion that USP36 acts as a protease that liberates FUBI from eS30: (1) The depletion of USP36 by RNAi impairs the processing of both endogenous and ectopically expressed FUBI-eS30. (2) Downregulation of USP36 by CRISPRi recapitulates the accumulation of uncleaved FUBI-eS30 and rescue experiments showed that cleavage depends on the catalytic activity of USP36. (3) Recombinant USP36 but not a catalytically deficient mutant can cleave FUBI-eS30 in vitro. Together, these results support the direct involvement of USP36 in cellular FUBI-eS30 cleavage.

The observation that 40S subunits carrying uncleaved FUBI are impaired in entering the pool of elongating ribosomes would predict that loss of USP36 affects 40S biogenesis, translation, and thereby cell proliferation and survival. Indeed, genetic ablation of mouse *Usp36* is lethal, and embryos die at the preimplantation stage (*Fraile et al., 2018*). Whether this is due to a ribosome biogenesis defect or has other reasons is currently unclear. Interestingly, it has recently been reported that depletion of USP36 causes mild defects in pre-rRNA processing and reduced protein translation, explained by an effect of USP36 on the SUMOylation status of components of nucleolar snoRNPs (*Ryu et al., 2021*). In light of our data, the observed deficiencies in pre-rRNA processing and protein translation could of course also be rationalized by deficient FUBI processing. Clearly, it will be interesting to investigate in the future if there is a link between FUBI processing and changes in snoRNP metabolism. It should be noted though that in our hands, downregulation of USP36 turned out to be insufficient to cause obvious defects in the recycling of RBFs, unlike the non-cleavable FUBI-eS30 reporter constructs. However, the level of endogenous uncleaved FUBI-eS30 upon USP36 depletion is much lower than that of the expressed non-cleavable FUBI-eS30 constructs (*Figure 6D*), indicating that USP36 depletion did not result in sufficient levels of uncleaved FUBI-eS30 to trigger a dominant-negative effect. Of course, it is also possible that there are other proteases working redundantly with USP36 on FUBI-eS30 processing.

While processing of the ubiquitin fusion proteins has been postulated to occur during their translation or rapidly thereafter in the cytoplasm (*Bachmair et al., 1986*; *Grou et al., 2015*; *Lacombe et al., 2009*), newly synthesized FUBI-eS30 must translocate from the cytoplasm to the nucleus for cleavage by USP36, which is localized in nucleoli (*Endo et al., 2009b*; *Urbé et al., 2012*). In support of this view, we readily detected uncleaved FUBI-eS30 in cell lysates (*Figure 1C*, *Figure 6A*), indicating that FUBI-eS30 persists in its unprocessed form after translation. In agreement with a nucle(ol)ar site of eS30 integration, it accumulated in the nucleoplasm and nucleoli upon inhibition of nuclear steps of 40S subunit synthesis. During the intracellular transport of eS30, the folded FUBI moiety may serve as in cis chaperone, limiting aggregation of its unstructured and positively charged fusion partner eS30, akin to the suggested role of ubiquitin for Ub-RP fusions (*Finley et al., 1989*; *Lacombe et al., 2009*; *Martín-Villanueva et al., 2019*). A ULP confined to the nucleolus such as USP36 seems perfectly suited to prevent premature FUBI-eS30 processing, although the existence of redundant FUBI-specific ULPs in other cellular compartments cannot be excluded at this point.

## USP36 – a multifunctional enzyme

The in vitro processing assays revealed that USP36 can cleave both FUBI and ubiquitin fusion proteins. Cross-reactivity of DUBs in vitro has been reported before, for instance for human UCHL3 and USP21, which not only act on ubiquitin but also on the UBLs NEDD8 and ISG15, respectively (*Larsen et al., 1998*; *Wada et al., 1998*; *Ye et al., 2011*). Recombinant USP36 was active toward

both Ub-EGFP and FUBI-EGFP, which is consistent with the observation that USP36 can act as a DUB on a variety of substrates (see below). USP36 was also active towards the FUBI-eS30 fusion protein, albeit less potently. We assume that this is due to the unfolded nature of the positively charged eS30 as compared to the compact fold of the larger EGFP. Based on this reasoning, it is conceivable that cleavage of FUBI-eS30 by USP36 might be stimulated once the fusion protein is incorporated into nascent ribosomal subunits. In such a setting, the nascent 40S subunit would contribute to specificity of USP36 towards FUBI-eS30.

Previous studies have suggested that USP36 deubiquitylates and thereby prevents proteasomal degradation of several nucleolar factors modified with K48-linked polyubiquitin chains, such as Myc, FBL, NPM, or DHX33 (*Endo et al., 2009b*; *Fraile et al., 2018*; *Sun et al., 2015*). Together with our cell-based assays on FUBI-eS30 processing, these data indicate that USP36 may indeed be a bispecific enzyme in vivo. As for other DUBs, a key question is how USP36 is targeted to its different substrates. The activity of DUBs towards specific substrates can be regulated by binding partners that present the ubiquitinated substrate to the enzyme, most prominently by interacting WD40 repeat (WDR)-containing proteins (*Mevissen and Komander, 2017*). A DUB interactome study has identified the nucleolar proteins and potential small subunit processome components WDR3 (UTP12) and WDR36 (UTP21) as high-confidence candidate interactors of USP36 (*Sowa et al., 2009*). It remains to be seen whether they or other pre-40S-associated factors direct FUBI-eS30 processing, perhaps by guiding USP36 to FUBI-eS30 after its incorporation into an early 40S precursor.

Last but not least, it remains to be seen whether there is a function of FUBI as a ubiquitin-like factor once it is released from eS30 by USP36. If FUBI were indeed conjugated to protein partners in the many different cell types it is constantly generated in, then FUBI conjugation may well exploit the regular ubiquitin-directed machinery, in analogy to the use of the DUB USP36 for its deconjugation.

## Materials and methods

A list of materials is provided in Appendix key resources table.

### Molecular cloning

The coding sequences of FUBI-eS30 and ubiquitin were amplified from HeLa cDNA. FUBI-eS30 was cloned into the pcDNA5/FRT/TO/cStHA or nHASt vectors (St, tandem Strep-tag II; HA, hemagglutinin epitope; *Wyler et al., 2011*) using the BamHI and EcoRV or BamHI and XhoI restriction sites, respectively. FUBI-eS30 was cloned into the pQE30 vector (Qiagen) using the BamHI and HindIII restriction sites. BamHI- and HindIII-digested FUBI or ubiquitin PCR fragments were cloned into a modified pET-28b(+) vector (Novagen) allowing for expression of a His$_6$-tagged fusion protein with EGFP that was cloned into the HindIII and XhoI sites. USP36 was amplified from Flag-HA-USP36 (kind gift from Wade Harper, Addgene plasmid #22579, RRID:Addgene_22579) and cloned using the BamHI and EcoRV restriction sites of a pcDNA5/FRT/TO/nEGFP vector, which was generated by inserting EGFP into the KpnI and HindIII restriction sites of pcDNA5/FRT/TO (Invitrogen). For baculoviral expression of His$_{10}$-USP36-TEV-St (TEV, Tobacco Etch Virus protease cleavage site), BamHI- and NotI-digested USP36 PCR fragments were cloned into a pFastBac Dual vector (Invitrogen).

Point mutations of FUBI-eS30 and USP36 constructs were generated by site-directed QuikChange mutagenesis (Agilent Technologies, see oligonucleotide sequences in Appendix key resources table). Note that the EGFP-USP36 constructs contain additional silent mutations introducing an NdeI restriction site. USP36 CRISPRi constructs were generated by inserting primer annealed guides into the BsaI sites of the pC2Pi vector (*Boneberg et al., 2019*) encoding tagged, catalytically inactive Cas9(D10A,H840A) separated by a P2A site from a puromycin cassette. The control guide plasmid containing a gRNA targeting the *Danio rerio tia1l* gene has been described previously (*Boneberg et al., 2019*). USP36-targeting gRNA sequences (USP36i_g1: 5'-GCGGGCCGAAGGAG TCGCCA-3', USP36i_g2: 5'-GGGACGCTCAGGGAGAACGT-3') were chosen according to *Horlbeck et al., 2016*.

### Antibodies

The following commercial antibodies were used in this study: anti-β-actin (Santa Cruz Biotechnology, sc-47778, RRID:AB_626632), anti-EIF1AD (Proteintech, 20528–1-AP, RRID:AB_10693533), anti-eS26

(Abcam, ab104050), anti-FAU (Abcam, ab135765), anti-HA (Enzo Life Sciences, ENZ-ABS120-0200), anti-His (Sigma-Aldrich, H1029, RRID:AB_260015), anti-Strep (IBA GmbH, 2-1507-001, RRID:AB_513133), anti-USP10 (Sigma-Aldrich, HPA006731, RRID:AB_1080495), anti-USP16 (Bethyl Laboratories, A301-615A, RRID:AB_1211387), and anti-USP36 (Sigma-Aldrich, HPA012082, RRID:AB_1858682). Secondary antibodies were purchased from Thermo Fisher Scientific: goat anti-mouse Alexa Fluor 594 (A-11005, RRID:AB_2534073) and goat anti-rabbit Alexa Fluor 488 (A-11008, RRID:AB_143165) were used for immunofluorescence analysis, goat anti-mouse Alexa Fluor Plus 680 (A32729, RRID:AB_2633278) and goat anti-rabbit Alexa Fluor Plus 800 (A32735, RRID:AB_2633284) were used for immunoblot analysis.

Antibodies against C21orf70, ENP1, eS1 (RPS3A), LSG1, LTV1, NMD3, NOB1, NOC4L, PNO1 (DIM2), RIOK1, RIOK2, RRP12, TSR1, uL23 (RPL23A), and uS3 (RPS3) have been described previously (*Montellese et al., 2020*; *Widmann et al., 2012*; *Wyler et al., 2014*; *Wyler et al., 2011*; *Zemp et al., 2014*; *Zemp et al., 2009*). Anti-AAMP, anti-FUBI, and anti-TRIP4 antibodies were raised in rabbits against affinity purified N-terminally hexahistidine-tagged full-length proteins and purified with the respective antigens coupled to SulfoLink resin (Thermo Fisher Scientific).

## Cell culture, cell lines, and treatments

Human cell lines were cultured in DMEM containing 10% fetal calf serum and 100 µg/ml penicillin/streptomycin (DMEM+/+) at 37°C and 5% $CO_2$. HeLa K cells were a kind gift from Dr. D. Gerlich (IMBA, Vienna, Austria). Tetracycline-inducible cell lines were generated by stable integration of pcDNA5-based constructs into the FRT site upon co-transfection with pOG44 plasmid (Invitrogen). HeLa FRT cells (*Häfner et al., 2014*) were used to obtain monoclonal FUBI-eS30 cell lines after selection with 0.4 mg/ml hygromycin B. A different parental HeLa FRT cell line (kind gift from Prof. M. Beck, EMBL, Heidelberg, Germany) was used for the generation of monoclonal USP36 cell lines after selection with 0.3 mg/ml hygromycin B. Polyclonal HEK293 Flp-In T-REx cell lines (Invitrogen) were generated as described previously (*Wyler et al., 2011*) by selection with 0.1 mg/ml hygromycin B and 15 µg/ml blasticidin S. The HEK293 HASt-GFP and HASt-C21orf70 cell lines have been described previously (*Wyler et al., 2011*; *Zemp et al., 2014*). Cell lines used in this study were not further authenticated after obtaining them from the indicated sources. All cell lines were tested negative for mycoplasma using PCR-based testing and they are not listed as commonly misidentified cell lines by the International Cell Line Authentication Committee.

Unless stated otherwise, HeLa FUBI-eS30-StHA WT and mutant cells were induced for 17 hr prior to harvest by the addition of 0.5 µg/ml and 0.1 µg/ml tetracycline (tet), respectively. Similarly, HEK293 FUBI-eS30 WT (N-, C-terminally tagged) and mutant (AA, GV) FUBI-eS30-StHA cells were induced for 24 hr prior to harvest with 0.5 µg/ml and 25 ng/ml tet, respectively. HEK293 HASt-GFP and HASt-C21orf70 cells were induced with 0.5 µg/ml tet for 24 hr prior to harvest. HeLa EGFP-USP36 WT and CA cell lines were induced for 96 hr with 3.125 ng/ml and 50 ng/ml tet, respectively.

Cycloheximide (CHX; cat# C7698) was purchased from Sigma-Aldrich, leptomycin B (LMB; cat#L-6100) was purchased from LC Laboratories, and tetracycline (tet; cat#550205) was purchased from Invitrogen.

## Immunoblot analysis

Samples in SDS sample buffer were separated on SDS-PAGE gels and proteins were transferred to nitrocellulose or PVDF membranes by semi-dry or wet blotting, respectively. Subsequently, membranes were blocked in 4% milk in 0.1% Tween-20 in PBS (PBST) and incubated with primary antibody diluted in 4% milk/PBST. After three washes for 5 min in PBST, membranes were incubated in secondary antibody diluted in 4% milk/PBST. After three washes for 5 min in PBST, signals were detected by an Odyssey (LI-COR) imaging system, measured with the Image Studio software (LI-COR) and quantified with the GraphPad Prism software.

## Sucrose density gradient analysis

For 10–45% sucrose gradients, protein expression in HEK293 cells was induced for 17 hr with 0.1 µg/ml tet before translation was arrested by treatment with 100 µg/ml CHX for 12 min at 37°C. Cells were lysed in 50 mM HEPES pH 7.5, 100 mM KCl, 3 mM $MgCl_2$, 0.5% (w/v) NP-40, 50 µg/ml CHX, 1 mM DTT, and protease inhibitors. The lysate was cleared by centrifugation (16,000 g, 5 min, 4°C)

and supernatant containing 600 µg total protein was loaded onto a linear 10–45% sucrose gradient in 50 mM HEPES pH 7.5, 100 mM KCl, 3 mM MgCl$_2$. After centrifugation (55,000 rpm, 60 min, 4°C) in a TLS55 rotor (Beckman Coulter), fractions were collected manually for precipitation with trichloro-acetic acid (TCA) and subsequent immunoblot analysis.

For 15–45% sucrose gradients, protein expression in HeLa cells was induced for 24 hr before cells were treated with 100 µg/ml CHX for 3 min at 37°C. Cells were lysed in 10 mM Tris pH 7.5, 100 mM KCl, 10 mM MgCl$_2$, 1% (w/v) TX-100, 100 µg/ml CHX, 1 mM DTT, and protease inhibitors. After centrifugation (10,000 g, 3 min, 4°C), cleared lysate containing 1.5 mg total protein was loaded onto a linear 15–45% sucrose gradient in 50 mM HEPES pH 7.5, 100 mM KCl, 10 mM MgCl$_2$. After centrifugation (38,000 rpm, 210 min, 4°C) in an SW41 rotor (Beckman Coulter), RNA content was measured at A$_{254}$ while fractions were collected using a Foxy Jr. Fraction Collector (ISCO) for subsequent TCA precipitation and immunoblot analysis. The peak area under the A$_{254}$ trace was measured with the Fiji software (*Schindelin et al., 2012*) and quantified using the GraphPad Prism software.

## Northern blot analysis

Total RNA was extracted from human cells with an RNeasy Mini kit (Qiagen). Northern blot was performed as described previously (*Tafforeau et al., 2013*). RNA (3 µg) was separated in an agarose-formaldehyde gel by electrophoresis (75 V, 5 hr) in 50 mM HEPES pH 7.8 containing 1 mM EDTA and stained with GelRed (Biotium, cat#41003). After washing the gel with 75 mM NaOH for 15 min, with 1.5 M NaCl in 0.5 M Tris pH 7 for 20 min, and 10X SSC (0.15 M tri-sodium citrate dihydrate pH 7.0, 1.5 M NaCl) for 15 min, the RNA was blotted onto a nylon membrane (Hybond-N$^+$, GE Healthcare) by capillary transfer. After UV crosslinking, the membrane was prehybridized in 50% (v/v) form-amide, 5X SSPE, 5X Denhardt's solution, 1% SDS, 200 µg/ml DNA (Roche, cat#11467140001) for 1 hr at 65°C. rRNA precursors were hybridized with radioactively ($^{32}$P) labeled probes (5'ITS1: 5'- CCTCGCCCTCCGGGCTCCGTTAATGATC-3'; ITS2: 5'- GCGCGACGGCGGACGACACCGCGGCGTC-3'; *Rouquette et al., 2005*) for 1 hr at 65°C and subsequent overnight incubation at 37°C. After washing the membrane three times for 5 min with 2X SSC (30 mM tri-sodium citrate dihydrate pH 7.0, 300 mM NaCl) at 37°C, a phosphor screen was exposed with the membrane and was subsequently scanned using a Typhoon FLA 9000 (GE Healthcare). Signals were measured with the Fiji software (*Schindelin et al., 2012*) and quantified with the GraphPad Prism software.

## Fluorescence in situ hybridization

FISH was performed as described previously (*Rouquette et al., 2005*) using a 5' Cy3-labeled 5'ITS1 probe (5'-CCTCGCCCTCCGGGCTCCGTTAATGATC-3'; Microsynth).

Cells grown on coverslips were fixed in 4% PFA/PBS for 30 min at room temperature. After two rinses with PBS, cells were permeabilized by overnight incubation with 70% ethanol. Cells were rehydrated by two incubations for 5 min in 10% formamide in 2X SSC (30 mM tri-sodium citrate dihydrate pH 7.0, 300 mM NaCl). Hybridization was performed for 4 hr at 37°C in 10% formamide in 2X SSC supplemented with 10% dextran sulfate, 10 mM ribonucleoside vanadyl complexes (Sigma-Aldrich, cat#R-3380), 50 µg/ml BSA (Sigma-Aldrich, cat#B-2518), 0.5 µg/µl tRNA (Sigma-Aldrich, cat#R-1753), and 0.5 ng/µl Cy3-5'ITS1 probe. Subsequently, cells were washed twice for 30 min with 10% formamide in 2X SSC before washing for 5 min with PBS and mounting of the coverslips using Vectashield (Vector Laboratories). After imaging by confocal microscopy, the cytoplasmic signal was measured with CellProfiler version 4.2.0 (*McQuin et al., 2018*) and quantified with the GraphPad Prism software.

## Immunofluorescence analysis

Immunofluorescence analysis was performed as described previously (*Zemp et al., 2009*). Cells grown on coverslips were fixed in 4% paraformaldehyde in PBS (PFA/PBS) for 20 min at room temperature and permeabilized in 0.1% Triton X-100 (TX-100) and 0.02% SDS in PBS for 5 min. Cells were blocked by incubation with 2% BSA/PBS for 10 min and blocking solution (10% goat serum in 2% BSA/PBS) for 30 min. Primary antibodies diluted in blocking solution were added for 1 hr and cover slips were washed three times for 5 min with 2% BSA/PBS. Cells were incubated with secondary antibodies for 30 min and washed three times for 5 min with 2% BSA/PBS. After a brief rinse with 0.1% TX-100% and 0.02% SDS in PBS, cells were incubated with 4% PFA/PBS supplemented

with 1 μg/ml Hoechst (Sigma-Aldrich, cat#63493) for 10 min. After washing with PBS, coverslips were mounted onto glass slides with Vectashield (Vector Laboratories) for confocal microscopy.

## Confocal microscopy

Images were acquired at an LSM 780 or 880 confocal microscope (Zeiss) equipped with a 63X, 1.4 NA oil DIC Plan-Apochromat objective.

## RNA interference

HeLa K cells were transfected with siRNA oligonucleotides using Opti-MEM medium and INTER-FERin (Polyplus transfection), whereas HeLa FRT and HEK293 cells were transfected using Opti-MEM and Lipofectamine RNAiMAX (Invitrogen).

The following siRNAs were used: si-AAMP (AAMP (5'-CAGGAUGGCAGCUUGAUCCUA-3', Microsynth), si-control (Allstars Negative Control siRNA; Qiagen, cat# 1027281), si-eS4X (5'-C UGGAGGUGCUAACCUAGGAA-3', Qiagen), si-FAU (5'-CCGGCGCUUUGUCAACGUUGU-3', Qiagen), si-uS19 (5'-UCACCUACAAGCCCGUAAA-3', Microsynth), si-USP10-1 (5'-UCGCUUUGGA UGGAAGUUCUA-3', Qiagen), si-USP10-2 (5'-UACGUCAACACCCAUGAUAGA-3', Qiagen), si-USP10-3 (5'-AACACAGCUUCUGUUGACUCU-3', Qiagen), si-USP10-4 (5'-AAGAACUAGUUCUUAC UUCAA-3'; Qiagen), si-USP36-1 (5'-CAAGAGCGUCUCGGACACCUA-3'; Qiagen, Microsynth), si-USP36-2 (5'-UCCGUAUAUGUCCCAGAAUAA-3'; Qiagen), si-USP36-3 (5'-CCGCAUCGAGAUGCCA UGCAU-3'; Qiagen), si-USP36-4 (5'-UUCCUUGUGAGUAGCUCUCAA-3'; Qiagen), si-XPO1 (5'-UG UGGUGAAUUGCUUAUAC-3')).

## CRISPR interference rescue experiments

Expression of the rescue constructs in HeLa cell lines was initiated 24 hr prior to CRISPRi plasmid transfection by the addition of tetracycline. Subsequently, cells were kept in medium supplemented with tetracycline. CRISPRi was performed as described previously (*Boneberg et al., 2019*) by transfecting the cells with pC2Pi plasmids carrying a catalytic dead dCas9 (Cas9(D10A,H840A)) fused to a gene-silencing KRAB domain and enabling transcription of a single guide RNA (sgRNA) from a U6 promoter. Cells were transfected with the plasmid using jetPRIME (Polyplus Transfection) and after 24 hr, cells were reseeded and selected for successful transfection by treatment with 2 μg/ml puromycin for 24 hr. Subsequently, the cells were washed twice with PBS and cultured in DMEM+/+ for 24 hr prior to harvest.

## StrepTactin pull-down

For StrepTactin affinity purification of HASt-C21orf70, HEK293 cells expressing the bait protein were lysed in 1 ml lysis buffer (10 mM Tris pH 7.5, 100 mM KCl, 2 mM $MgCl_2$, 1 mM DTT, 0.5% (w/v) NP-40, containing protease and phosphatase inhibitors) per ~3 x $10^7$ cells using a dounce homogenizer. Lysates were cleared by centrifugation (4'500 g, 12 min, 4°C) before incubation with pre-equilibrated StrepTactin sepharose beads (IBA, cat#2-1201-025) for 60 min at 4°C while rotating. Beads were then washed twice in lysis buffer, twice with 10 mM Tris pH 7.5, 100 mM KCl, 2 mM $MgCl_2$, and eluted by incubation in the same buffer supplemented with 2 mM biotin for 4 min on ice, to which SDS-PAGE sample buffer was added post elution.

## Affinity purification mass spectrometry analysis

StrepTactin affinity purification was performed as described previously (*Wyler et al., 2011*). Extracts of HEK293 cells expressing the bait protein of interest were prepared in 1 ml lysis buffer (10 mM Tris pH 7.5, 100 mM KCl, 2 mM $MgCl_2$, 1 mM DTT, 0.5% (w/v) NP-40, 2 μM avidin, containing protease and phosphatase inhibitors) per ~3 x $10^7$ cells using a dounce homogenizer. Lysates were cleared by centrifugation (4'500 g, 12 min, 4°C) before incubation with pre-equilibrated StrepTactin sepharose beads (IBA, cat#2-1201-025) for 30 min at 4°C while rotating. Beads were then washed six times with 10 mM Tris pH 7.5, 100 mM KCl, 2 mM $MgCl_2$, containing phosphatase inhibitors and eluted by incubation in the same buffer supplemented with 2 mM biotin for 4 min on ice. Eluted proteins were precipitated with TCA, denatured with 6 M urea, reduced with 12 mM DTT for 30 min at 32°C, and alkylated with 40 mM iodoacetamide for 45 min at 25°C in the dark. The sample was diluted 1:7 with 0.1 M $NH_4HCO_3$ before overnight digestion of proteins with trypsin (Promega, cat#V5113) at 30°C.

Digestion was stopped by the addition of 2% (v/v) formic acid before peptides were cleaned using Pierce C18 spin columns (Thermo Fisher Scientific, cat#89870) according to the manufacturer's protocol. SpeedVac-dried peptides were taken up in 0.1% (v/v) formic acid at a concentration of 1 µg/µl.

Peptides (1 µg/run) were separated with an EASY-nLC 1000 nanoflow liquid chromatograph (Thermo Fisher) on a 75 µm diameter, 40 cm long new Objective emitter packed with ReproSil Gold 120 C18 resin (1.9 µm, Dr. Maisch) and eluted at 300 nl/min with a linear gradient of 5–35% Buffer A (0.1% (v/v) formic acid in water) in Buffer B (0.1% (v/v) formic acid in acetonitrile) for 90 min. MS/MS measurements using data-dependent acquisition (top 20, excluding the singly and unassigned charged species, 30 s dynamic exclusion) were carried out on a Q Exactive Plus mass spectrometer (Thermo Fisher) equipped with a Nanospray Flex ion source and acquired with Xcalibur version 4.2.28.14. Data analysis, using a human Uniprot-reviewed database supplemented with the protein sequences of HASt-GFP and one sequence each for the WT, AA, and GV FUBI-eS30-StHA bait proteins, was performed using the Comet search engine as described previously (*Montellese et al., 2020*). The spectral counts of the confidently identified interactors (SAINT Bayesian false discovery rate < 1%) were normalized to the protein length in amino acids (UniProt database) after addition of a pseudocount of 0.1. The enrichment of the average normalized spectral counts identified with the mutants (AA + GV) compared to WT FUBI-eS30-StHA were tested by ANOVA.

## Protein expression and purification

Substrates for in vitro processing assays were expressed in *E. coli* and purified by Nickel affinity chromatography (Ni$^{2+}$-NTA-sepharose resin, Qiagen). His$_6$-FUBI-eS30 WT and AA constructs were expressed in *E. coli* BLR(pREP4) cells (Qiagen) cultured in DYT medium (1% tryptone, 1% yeast extract, 0.5% NaCl) for 4 hr at 25°C or overnight at 18°C, respectively, after induction with 0.5 mM IPTG (isopropyl-β-D-thiogalactopyranoside). His$_6$-FUBI-EGFP and His$_6$-Ub-EGFP were expressed in *E. coli* Rosetta (DE3) cells (Novagen) cultured in DYT medium for 4 hr at 25°C after induction with 0.5 mM IPTG. Cells were resuspended in lysis buffer (~30 ml/l culture; 50 mM Tris pH 7.5, 700 mM NaCl, 3 mM MgCl$_2$, 5% (v/v) glycerol, 50 mM imidazole, 2 mM β-mercapto-ethanol) supplemented with 5 µg/ml DNase I (Roche) and lysed using a homogenizer (Avestin Emulsiflex C5, ATA Scientific; passed three times, 4°C). Cell lysates were cleared by centrifugation (55,000 rpm, 60 min, 4°C, 70 Ti rotor (Beckman Coulter)), added to Ni-NTA beads (5 ml slurry/l culture) pre-equilibrated in lysis buffer, and incubated for at least 1 hr at 4°C while rotating. Bound protein was washed with lysis buffer and eluted with elution buffer (30 mM Tris pH 7.5, 420 mM NaCl, 1.8 mM MgCl$_2$, 3% (v/v) glycerol, 400 mM imidazole). Purified proteins were were concentrated > 60 µM using Amicon Ultra centrifugal filters (Merck; MWCO 3 kDa), rebuffered to 50 mM Tris pH 7.5, 200 mM NaCl, 3 mM MgCl$_2$, 5% (v/v) glycerol, 5 mM DTT using NAP-5 or PD-10 desalting columns (Amersham) and flash frozen with liquid nitrogen for storage at −80°C.

USP36 enzymes for in vitro processing assays were expressed in Sf9 insect cells using the Bac-to-Bac system (Invitrogen) and purified by StrepTactin affinity chromatography (IBA Lifesciences). To generate baculoviruses, Sf9 cells (CSL) cultured in SF-900 II SFM at 27°C were transfected at a density of 0.4 x 10$^6$ ml$^{-1}$ with 1 µg bacmid DNA using Excort IV Transfection Reagent (Merck) according to the manufacturer's protocol. Sf9 cells were infected with P1 virus (50 ml/l) at a density of 1.8 x 10$^6$ ml$^{-1}$. After 48 hr, Sf9 cells were resuspended in lysis buffer (~50 ml/l culture; 50 mM Tris pH 7.5, 700 mM NaCl, 3 mM MgCl$_2$, 2 mM DTT, 0.4 mg/ml PMSF, 10 µg/ml aprotinin, 10 µg/ml leupeptin, 1 µg/ml pepstatin A, 5 µg/ml DNase I, 2 µM avidin) and lysed using a homogenizer (Avestin Emulsiflex C5, ATA Scientific, passed three times, 4°C). Cell lysates were cleared by centrifugation (50,000 rpm, 60 min, 4°C, 70 Ti rotor (Beckman Coulter)), added to StrepTactin beads (1.5 ml slurry/l culture) pre-equilibrated in wash buffer (50 mM Tris pH 7.5, 700 mM NaCl, 3 mM MgCl$_2$, 2 mM DTT), and incubated for 1 hr at 4°C while rotating. The beads were washed with wash buffer and eluted with 2 mM biotin in wash buffer. Enzymes were concentrated to ~40 µM using Amicon Ultra centrifugal filters (Merck; MWCO 10 kDa) and flash frozen with liquid nitrogen for storage at −80°C.

## In vitro processing assays

Purified substrates and enzymes were centrifuged (16,000 g, 5 min, 4°C) before protein concentration was determined by Bradford assay (Bio-Rad) according to the manufacturer's instructions. For reduction of (active site) cysteines, enzymes (0.5 µM) were incubated in assay buffer (50 mM Tris pH

7.5, 100 mM potassium acetate, 3 mM MgCl$_2$, 5 mM DTT, 5% (v/v) glycerol, 0.05% (w/v) Tween-20) for 5 min at 37°C prior to addition of substrates (2.5 µM) and immediate mixing by pipetting. Samples taken were mixed with 2X SDS sample buffer and heated for 1 min at 95°C for analysis on Coomassie brilliant blue R250-stained 4–12% SDS-PAGE gels (Thermo Fisher Scientific). Signals were measured with the Fiji software (*Schindelin et al., 2012*) and quantified with the GraphPad Prism software.

## Acknowledgements

We thank Dr. C Montellese and C Gafko for critical reading of the manuscript, and the other members of the Kutay lab for helpful discussions. We are very grateful to Dr. P Boersema and Prof. P Picotti whose pilot MS analysis initiated our work on USP36. Microscopy was performed on instruments of the ETHZ Microscopy Center Scope M. This work was initially funded by a grant of the Swiss National Science Foundation (SNSF) to U.K. (31003A_166565), and subsequently by the NCCR 'RNA and disease'.

## Additional information

### Funding

| Funder | Grant reference number | Author |
| --- | --- | --- |
| Swiss National Science Foundation | 31003A_166565 | Ulrike Kutay |
| Swiss National Science Foundation | NCCR RNA and Disease | Ulrike Kutay |

The funders had no role in study design, data collection and interpretation, or the decision to submit the work for publication.

### Author contributions

Jasmin van den Heuvel, Conceptualization, Formal analysis, Investigation, Visualization, Writing - original draft, Writing - review and editing; Caroline Ashiono, Emanuel Wyler, Investigation; Ludovic C Gillet, Data curation, Formal analysis, Investigation; Kerstin Dörner, Formal analysis; Ivo Zemp, Conceptualization, Supervision, Investigation, Writing - original draft, Writing - review and editing; Ulrike Kutay, Conceptualization, Data curation, Supervision, Funding acquisition, Writing - original draft, Project administration, Writing - review and editing

### Author ORCIDs

Ludovic C Gillet http://orcid.org/0000-0002-1001-3265
Emanuel Wyler http://orcid.org/0000-0002-9884-1806
Ivo Zemp https://orcid.org/0000-0003-1549-9871
Ulrike Kutay https://orcid.org/0000-0002-8257-7465

### Decision letter and Author response

Decision letter https://doi.org/10.7554/eLife.70560.sa1
Author response https://doi.org/10.7554/eLife.70560.sa2

## Additional files

### Supplementary files

• Supplementary file 1. Proteomic analysis of the interactome of WT and non-cleavable FUBI-eS30-StHA. (S2-1) Spectral counts and SAINT Bayesian false discovery rates (BFDRs) of proteins identified in affinity purification mass spectrometry experiments of HASt-GFP (GFP), FUBI-eS30-StHA wild-type (WT), G73,74A (AA) or G74V (GV) performed in three independent biological replicates. (S2-2) Spectral counts of the confidently identified interactors with a BFDR < 0.01 for at least one bait were normalized to protein length (per 1000 amino acids, UniProt database) after addition of a pseudocount

of 0.1. The average spectral counts identified on the individual (AA, GV) and combined (mut) non-cleavable mutants compared to WT FUBI-eS30-StHA were determined as a $\log_2$ fold change ($\log_2$FC). The enrichment of interactors on mut vs. WT was tested by ANOVA (p value and adjusted p values using FDR correction). Significantly enriched interactors ($|\log_2$FC$| >$ one with an adjusted p value $< 0.05$) were categorized as deubiquitinase (DUB), 40S ribosome biogenesis factor (RBF), 60S RBF, ribosome-associated, or other enriched interactor.

- Transparent reporting form

### Data availability

Source data have been provided for Figure 5.

The following dataset was generated:

| Author(s) | Year | Dataset title | Dataset URL | Database and Identifier |
|---|---|---|---|---|
| Kutay U | 2020 | AP-MS analysis of human FUBIeS30/FAU WT and non-cleavable G73,4A and G74V mutants | https://www.ebi.ac.uk/pride/archive/projects/PXD021864 | PRIDE, PXD021864 |

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

# Appendix 1

**Appendix 1—key resources table** .

| Reagent type (species) or resource | Designation | Source or reference | Identifiers | Additional information |
|---|---|---|---|---|
| Cell line (*Homo sapiens*) | HEK293 Flp-In T-REx | Thermo Fisher Scientific | Cat# R78007 RRID:CVCL_U427 | |
| Cell line (*Homo sapiens*) | HEK293 Flp-In T-REx FUBI-eS30-StHA | This paper | | See Materials and methods, *Cell culture, cell lines, and treatments* |
| Cell line (*Homo sapiens*) | HEK293 Flp-In T-REx FUBI(AA)-eS30-StHA | This paper | | AA: G73,74A See Materials and methods, *Cell culture, cell lines, and treatments* |
| Cell line (*Homo sapiens*) | HEK293 Flp-In T-REx FUBI(GV)-eS30-StHA | This paper | | GV: G74V See Materials and methods, *Cell culture, cell lines, and treatments* |
| Cell line (*Homo sapiens*) | HEK293 Flp-In T-REx HASt-C21orf70 | (*Zemp et al., 2014*) DOI: 10.1242/jcs.138719 | | |
| Cell line (*Homo sapiens*) | HEK293 Flp-In T-REx HASt-FUBI-eS30 | This paper | | See Materials and methods, *Cell culture, cell lines, and treatments* |
| Cell line (*Homo sapiens*) | HEK293 Flp-In T-REx HASt-GFP | (*Wyler et al., 2011*) DOI: 10.1261/rna.2325911 | | |
| Cell line (*Homo sapiens*) | HeLa Flp-In T-REx | (*Häfner et al., 2014*) DOI: 10.1038/ ncomms5397 | | Obtained from T. Mayer (University of Konstanz) |
| Cell line (*Homo sapiens*) | HeLa Flp-In T-REx FUBI-eS30-StHA | This paper | | See Materials and methods, *Cell culture, cell lines, and treatments* |
| Cell line (*Homo sapiens*) | HeLa Flp-In T-REx FUBI(AA)-eS30-StHA | This paper | | AA: G73,74A See Materials and methods, *Cell culture, cell lines, and treatments* |
| Cell line (*Homo sapiens*) | HeLa Flp-In T-REx FUBI(GV)-eS30-StHA | This paper | | GV: G74V See Materials and methods, *Cell culture, cell lines, and treatments* |
| Cell line (*Homo sapiens*) | HeLa FRT/TetR | Other | | Obtained from M. Beck (EMBL, Heidelberg) |

*Continued on next page*

*Appendix 1—key resources table continued*

| Reagent type (species) or resource | Designation | Source or reference | Identifiers | Additional information |
|---|---|---|---|---|
| Cell line (*Homo sapiens*) | HeLa FRT/TetR EGFP-USP36 | This paper | | See Materials and methods, *Cell culture, cell lines, and treatments* |
| Cell line (*Homo sapiens*) | HeLa FRT/TetR EGFP-USP36(CA) | This paper | | CA: C131A See Materials and methods, *Cell culture, cell lines, and treatments* |
| Cell line (*Homo sapiens*) | HeLa Kyoto, HeLa K | Other | RRID:CVCL_1922 | Obtained from D. Gerlich (IMBA, Vienna) |
| Antibody | anti-AAMP (rabbit polyclonal) | This paper | | WB (1:800) See Materials and methods, *Antibodies* |
| Antibody | anti-β-actin (mouse monoclonal) | Santa Cruz Biotechnology | Cat# sc-47778 RRID:AB_626632 | WB (1:1000) |
| Antibody | anti-C21ORF70 (rabbit polyclonal) | (*Montellese et al., 2020*) DOI: 10.1093/nar/gkx253 | | WB (1:500) |
| Antibody | anti-EIF1AD | Proteintech | Cat# 20528-1-AP RRID:AB_10693533 | WB (1:1000) |
| Antibody | anti-ENP1 (rabbit polyclonal) | (*Zemp et al., 2009*) DOI: 10.1083/jcb.200904048 | | IF (1:15,000) WB (1:1000) |
| Antibody | anti-eS1 (RPS3A) (rabbit polyclonal) | (*Wyler et al., 2011*) DOI: 10.1261/rna.2325911 | | WB (1:1000) |
| Antibody | anti-eS26 (RPS26) (rabbit polyclonal) | Abcam | Cat# ab104050 RRID:AB_10710999 | WB (1:500) |
| Antibody | anti-FAU (FUBI-eS30) (rabbit polyclonal) | Abcam | Cat# ab135765 | WB (1:500) |
| Antibody | anti-FUBI (rabbit polyclonal) | This paper | | WB (1:2000) See Materials and methods, *Antibodies* |
| Antibody | anti-HA (mouse monoclonal) | Enzo Life Sciences | ENZ-ABS120-0200 | IF (1:2000) WB (1:1000) |
| Antibody | anti-His (mouse monoclonal) | Sigma-Aldrich | Cat# H1029 RRID:AB_260015 | WB (1:2000) |
| Antibody | anti-LSG1 (rabbit polyclonal) | (*Wyler et al., 2014*) DOI: 10.1016/j.febslet.2014.08.013 | | WB (1:3000) |
| Antibody | anti-LTV1 (rabbit polyclonal) | (*Zemp et al., 2009*) DOI: 10.1083/jcb.200904048 | | IF (1:4000) WB (1:2000) |
| Antibody | anti-NMD3 (rabbit polyclonal) | (*Zemp et al., 2009*) DOI: 10.1083/jcb.200904048 | | IF (1:1000) WB (1:10,000) |
| Antibody | anti-NOB1 (rabbit polyclonal) | (*Zemp et al., 2009*) DOI: 10.1083/jcb.200904048 | | IF (1:5000) WB (1:2000) |

*Continued on next page*

*Appendix 1—key resources table continued*

| Reagent type (species) or resource | Designation | Source or reference | Identifiers | Additional information |
|---|---|---|---|---|
| Antibody | anti-NOC4L (rabbit polyclonal) | (*Wyler et al., 2011*) DOI: 10.1261/rna.2325911 | | WB (1:5000) |
| Antibody | anti-PNO1 (DIM2) (rabbit polyclonal) | (*Zemp et al., 2009*) DOI: 10.1083/jcb.200904048 | | IF (1:2000) WB (1:2000) |
| Antibody | anti-RIOK1 (rabbit polyclonal) | (*Widmann et al., 2012*) DOI: 10.1091/mbc.E11-07-0639 | | IF (1:8000) WB (1:1000) |
| Antibody | anti-RIOK2 (rabbit polyclonal) | (*Zemp et al., 2009*) DOI: 10.1083/jcb.200904048 | | IF (1:5000) WB (1:5000) |
| Antibody | anti-RRP12 (rabbit polyclonal) | (*Wyler et al., 2011*) DOI: 10.1261/rna.2325911 | | IF (1:2000) WB (1:1000) |
| Antibody | anti-Strep (mouse monoclonal) | IBA GmbH | Cat# 2-1507-001 RRID:AB_513133 | WB (1:1000) |
| Antibody | anti-TRIP4 (rabbit polyclonal) | This paper | | WB (1:50,000) See Materials and methods, *Antibodies* |
| Antibody | anti-TSR1 (rabbit polyclonal) | ##(*Zemp et al., 2014*)# DOI: 10.1242/jcs.138719 | | WB (1:10,000) |
| Antibody | anti-uL23 (RPL23A) (rabbit polyclonal) | (*Wyler et al., 2011*) DOI: 10.1261/rna.2325911 | | WB (1:200) |
| Antibody | anti-uS3 (RPS3) (rabbit polyclonal) | (*Zemp et al., 2009*) DOI: 10.1083/jcb.200904048 | | WB (1:1000) |
| Antibody | anti-USP10 (rabbit polyclonal) | Sigma-Aldrich | Cat# HPA006731 RRID:AB_1080495 | WB (1:1000) |
| Antibody | anti-USP16 (rabbit polyclonal) | Bethyl Laboratories | Cat# A301-615A RRID:AB_1211387 | WB (1:500) |
| Antibody | anti-USP36 (rabbit polyclonal) | Sigma-Aldrich | Cat# HPA012082 RRID:AB_1858682 | IF (1:1000) WB (1:250) |
| Antibody | goat anti-mouse Alexa Fluor 594 | Thermo Fisher Scientific | A-11005, RRID:AB_2534073 | IF (1:250) |
| Antibody | goat anti-rabbit Alexa Fluor 488 | Thermo Fisher Scientific | A-11008, RRID:AB_143165 | IF (1:250) |
| Antibody | goat anti-mouse Alexa Fluor Plus 680 | Thermo Fisher Scientific | Cat# A32729 RRID:AB_2633278 | WB (1:10,000) |
| Antibody | goat anti-rabbit Alexa Fluor Plus 800 | Thermo Fisher Scientific | Cat# A32735, RRID:AB_2633284 | WB (1:10,000) |
| Recombinant DNA reagent | pC2Pi/control | (*Boneberg et al., 2019*) DOI: 10.1261/rna.069609.118 | | See Materials and methods, *Molecular cloning* |

*Appendix 1—key resources table continued*

| Reagent type (species) or resource | Designation | Source or reference | Identifiers | Additional information |
|---|---|---|---|---|
| Recombinant DNA reagent | pC2Pi/USP36i-g1 | This paper | | USP36 guide 1 See Materials and methods, *Molecular cloning* |
| Recombinant DNA reagent | pC2Pi/USP36i-g2 | This paper | | USP36 guide 2 See Materials and methods, *Molecular cloning* |
| Recombinant DNA reagent | pCDNA5/FRT/TO/FUBI-eS30-StHA | This paper | | See Materials and methods, *Molecular cloning* |
| Recombinant DNA reagent | pCDNA5/FRT/TO/FUBI(AA)-eS30-StHA | This paper | | AA: G73,74A See Materials and methods, *Molecular cloning* |
| Recombinant DNA reagent | pCDNA5/FRT/TO/FUBI(GV)-eS30-StHA | This paper | | GV: G74V See Materials and methods, *Molecular cloning* |
| Recombinant DNA reagent | pCDNA5/FRT/TO/EGFP-USP36 | This paper | | See Materials and methods, *Molecular cloning* |
| Recombinant DNA reagent | pCDNA5/FRT/TO/EGFP-USP36 (CA) | This paper | | CA: C131A See Materials and methods, *Molecular cloning* |
| Recombinant DNA reagent | pCDNA5/FRT/TO/HASt-FUBI-eS30 | This paper | | See Materials and methods, *Molecular cloning* |
| Recombinant DNA reagent | pDEST/LTR/Flag-HA-USP36 | Addgene (*Sowa et al., 2009*) DOI: 10.1016/j.cell.2009.04.042 | Cat# 22579 RRID: Addgene_22579 | See Materials and methods, *Molecular cloning* |
| Recombinant DNA reagent | pET-28b(+)/His$_6$-FUBI-EGFP | This paper | | See Materials and methods, *Molecular cloning* |
| Recombinant DNA reagent | pET-28b(+)/His$_6$-Ub-EGFP | This paper | | See Materials and methods, *Molecular cloning* |
| Recombinant DNA reagent | pFBD/EGFP/His$_{10}$-USP36-TEV-St | This paper | | pFBD: pFastBac Dual See Materials and methods, *Molecular cloning* |
| Recombinant DNA reagent | pFBD/EGFP/His$_{10}$-USP36(CA)-TEV-St | This paper | | pFBD: pFastBac Dual, CA: C131A See Materials and methods, *Molecular cloning* |
| Recombinant DNA reagent | pQE30/His$_6$-FUBI-eS30 | This paper | | See Materials and methods, *Molecular cloning* |
| Recombinant DNA reagent | pQE30/His$_6$-FUBI(AA)-eS30 | This paper | | AA: G73,74A See Materials and methods, *Molecular cloning* |

*Continued on next page*

*Appendix 1—key resources table continued*

| Reagent type (species) or resource | Designation | Source or reference | Identifiers | Additional information |
|---|---|---|---|---|
| Sequence-based reagent | QuikChange primers for FUBI (AA)- eS30 | Sigma-Aldrich | | 5'-GCAGGCCGC ATGCTTGc AGcTAAAGTTC ATGGTTCC-3', 5'-GGAACCATGAA CTTTAgC TgCAAGCATG CGGCCTGC-3' See Materials and methods, *Molecular cloning* |
| Sequence-based reagent | QuikChange primers for FUBI(GV)-eS30 | Sigma-Aldrich | | 5'-GGCCGCA TGCTTGGA GtTAAAGTTC ATGGTTCC-3', 5'-GGAACCATG AACTTTAaCT CCAAGCAT GCGGCC-3' See Materials and methods, *Molecular cloning* |
| Sequence-based reagent | QuikChange primers for USP36(CA) | Sigma-Aldrich | | 5'-CCACAACCTaG GCAACACC gcCTTTCTCAA TGCCACC-3', 5'-GGTGGCATTG AGAAAGgc GGTGTTGCCtA GGTTGTGG-3' See Materials and methods, *Molecular cloning* |
| Sequence-based reagent | QuikChange primers for USP36 (Ndel) | Sigma-Aldrich | | 5'- CCGTGTGCAAGA GCGTCagc GAtACaTAtGACCCC TACTTGGAC-3', 5'-GTCCAAGTAG GGGTCaTAtGT aTCgctGACGCTC TTGCACACGG-3' See Materials and methods, *Molecular cloning* |
| Sequence-based reagent | 5'ITS1 | Microsynth (*Rouquette et al., 2005*) DOI: 10.1038/sj.emboj. 7600752 | | 5'-CCTCGCCCTCCG GGCTCCGTTAATGATC- 3' See Materials and methods, *Northern blot analysis* |
| Sequence-based reagent | ITS2 | Microsynth (*Rouquette et al., 2005*) DOI: 10.1038/sj.emboj. 7600752 | | 5'- GCGCGACGGCGGAC GACACCGCGGCGTC- 3' See Materials and methods, *Northern blot analysis* |

*Continued on next page*

*Appendix 1—key resources table continued*

| Reagent type (species) or resource | Designation | Source or reference | Identifiers | Additional information |
|---|---|---|---|---|
| Sequence-based reagent | Cy3-5′ITS1 | Microsynth (*Rouquette et al., 2005*) DOI: 10.1038/sj.emboj.7600752 | | 5′-CCTCGCCCTCCGGG CTCCGTTAATGATC-3′ See Materials and methods, *Fluorescence* in situ *hybridization* |
| Sequence-based reagent | si-AAMP | Microsynth | | 5′-CAGGAUGGCAGC UUGAUCCUA-3 See Materials and methods, *RNA interference* |
| Sequence-based reagent | si-control | Qiagen | Cat# 1027281 | Allstars Negative Control siRNA See Materials and methods, *RNA interference* |
| Sequence-based reagent | si-eS4X (RPS4X) | Qiagen | | 5′-CUGGAGGUGCUA ACCUAGGAA-3′ See Materials and methods, *RNA interference* |
| Sequence-based reagent | si-FAU (FUBI-eS30) | Qiagen | | 5′-CCGGCGCUUUGUC AACGUUGU-3′ See Materials and methods, *RNA interference* |
| Sequence-based reagent | si-uS19 (RPS15) | Microsynth (*Rouquette et al., 2005*) DOI: 10.1038/sj.emboj.7600752 | | 5′-UCACCUACAAGC CCGUAAA-3′ See Materials and methods, *RNA interference* |
| Sequence-based reagent | si-USP10-1 | Qiagen | | 5′-UCGCUUUGGAU GGAAGUUCUA-3′ See Materials and methods, *RNA interference* |
| Sequence-based reagent | si-USP10-2 | Qiagen | | 5′-UACGUCAACACC CAUGAUAGA-3′ See Materials and methods, *RNA interference* |
| Sequence-based reagent | si-USP10-3 | Qiagen | | 5′-AACACAGCUUCU GUUGACUCU-3′ See Materials and methods, *RNA interference* |
| Sequence-based reagent | si-USP10-4 | Qiagen | | 5′-AAGAACUAGUU CUUACUUCAA-3′ See Materials and methods, *RNA interference* |
| Sequence-based reagent | si-USP36-1 | Qiagen, Microsynth | | 5′-CAAGAGCGUCU CGGACACCUA-3′ See Materials and methods, *RNA interference* |

*Continued on next page*

*Appendix 1—key resources table continued*

| Reagent type (species) or resource | Designation | Source or reference | Identifiers | Additional information |
|---|---|---|---|---|
| Sequence-based reagent | si-USP36-2 | Qiagen | | 5'-UCCGUAUAUGUC CCAGAAUAA-3' See Materials and methods, *RNA interference* |
| Sequence-based reagent | si-USP36-3 | Qiagen | | 5'-CCGCAUCGAGA UGCCAUGCAU-3' See Materials and methods, *RNA interference* |
| Sequence-based reagent | si-USP36-4 | Qiagen | | 5'-UUCCUUGUGAGU AGCUCUCAA-3' See Materials and methods, *RNA interference* |
| Sequence-based reagent | si-XPO1 (CRM1) | Microsynth (*Zemp et al., 2009*) DOI: 10.1083/jcb.200904048 | | 5'-UGUGGUGAAUUG CUUAUAC-3' See Materials and methods, *RNA interference* |
| Chemical compound, drug | cycloheximide, CHX | Sigma-Aldrich | Cat# C7698 | |
| Chemical compound, drug | Leptomycin B | LC Laboratories | Cat# L-6100 | |
| Chemical compound, drug | tetracycline, tet | Invitrogen | Cat# 550205 | |

