## [Decision Letter]

**Acceptance summary:**

This study has identified USP36 as the enzyme that processes FAU, a ribosomal protein precursor comprised of a fusion between ubiquitin-like protein FUBI and the ribosomal protein eS30. This is an important advance because correct processing of FAU is crucial for biogenesis of the 40S ribosomal subunit. Knowing the identity of the processing enzyme now opens this step in ribosome biogenesis to molecular and mechanistic analysis.

**Decision letter after peer review:**

Thank you for submitting your article "Processing of the Ribosomal Ubiquitin-Like Fusion Protein FUBI-eS30/FAU is Required for 40S Maturation and Depends on USP36" for consideration by *eLife*. Your article has been reviewed by 3 peer reviewers, one of whom is a member of our Board of Reviewing Editors, and the evaluation has been overseen by David Ron as the Senior Editor. The reviewers have opted to remain anonymous.

The reviewers have discussed their reviews with one another, and the Reviewing Editor has drafted this to help you prepare a revised submission. All three reviewers agreed that this was important work that adds to our understanding of ribosome biogenesis. Two of the reviewers thought that some of the specific conclusions need clarification. Most of these comments can be addressed by changing the text and/or tempering certain statements. We've included the individual reviews below. Addressing some of the individual points with experiments will likely strengthen the paper but are not necessary for acceptance of the revised manuscript. We look forward to receiving your revised manuscript.

Essential revisions:

1) Over-expression of the uncleavable form of FUBI-eS30 results in several dominant negative phenotypes. This transgene could be acting as a sink for a variety of factors needed for ribosome biogenesis, in either a specific or non-specific manner. At the very least, this caveat should be mentioned/discussed.

2) In Figures 3 and 4, imaging data is used to infer differences in ribosome biogenesis and/or association of certain factors with the pre-40S. Without biochemical experiments, the authors should temper their conclusions in the text regarding these data.

3) Figure 7 and associated text: The authors make statements regarding the kinetics of the reaction; however, they do not quantify these results. Please quantify the kinetics to support these claims.

4) To support USP36 as being the main protease for FUBI-eS30, the authors cite Rye et al. 2021. Although that reference shows USP36's involvement in pre-rRNA processing and translation, it is unclear if their results yielded a similar pre-40S defect as in this manuscript. While the involvement of USP36 in processing FUBI-eS30 is fairly clear, the authors should comment on the possibility that other DUBs may participate in this process.

5) Pg 19, paragraph 2: The authors' state "uncleaved FUBI-eS30 USP36 depletion was at least ten times lower compared to the expressed non-cleavable FUBI-eS30 constructs". Please reference the figure or present the data to support this statement as this is an important aspect of the experimental design for the reader to know.

*Reviewer #1 (Recommendations for the authors):*

1) Ribosome biogenesis assays should be performed to test whether disruption of USP36 results in similar defects as expression of the uncleavable FUBI-eS30 constructs.

2) Over-expression of the uncleavable form of FUBI-eS30 results in several dominant negative phenotypes. This transgene could be acting as a sink for a variety of factors needed for ribosome biogenesis. At the very least, this caveat should be mentioned and elaborated on in the discussion. Do these cells grow at different rates versus the wild-type controls and/or parental lines? If so, these changes could be further feeding back onto ribosome biogenesis. Does altering the level of transgene expression change any of the described phenotypes. These points are important in regards to some of the conclusions made in the paper (e.g. page 10-" In summary, the comprehensive examination of the (re)localization of various RBFs indicates that FUBI-eS30 cleavage is a prerequisite for recycling of the RBFs RIOK2, PNO1, NOB1, and RIOK1 in final steps of cytoplasmic 40S maturation."

3) Figure 1C is a bit unconvincing. Are the endogenous protein levels unaltered by expression of the transgenes? Can the authors provide some quantification?

4) Imaging data in Figure 3 is used to infer that eS30 joins pre-40S ribosome subunits in the nucleus. Pulldown of eS30 with nuclear pre-40S is needed to strengthen this conclusion.

5) On Page 9, the authors state "Strikingly, PNO1, an RBF that localizes to nucleoli under steady state conditions, relocated to the cytoplasm. Similarly, a prominent recycling effect was also evident for RIOK2 and NOB1. Both are cytoplasmic at steady state but in cells expressing non-cleavable FUBI-eS30-StHA, they failed to accumulate in the nucleus upon LMB treatment, reflecting a failure in their timely release from cytoplasmic pre-40S subunits (Zemp et al., 2009)." Biochemical experiments are needed to confirm association/ dissociation with the ribosome.

6) Typo on page 47- brillinat should be brilliant

*Reviewer #2 (Recommendations for the authors):*

1. In situ hybridization and immunofluorescence images do not have a nuclear and/or nucleolar staining associated with them, but claims are made to the localization and abundance of the fluorescence within these cellular compartments. Please indicate with the appropriate nuclear and/or nucleolar staining markers. In the Materials and methods Section it was reported that Hoechst was used, please report those images for the nuclear marker. ENP1 seems to serve as a nucleolar marker since co-stained with anti-HA (Figure 3). Additionally, nuclear vs cytoplasmic signal quantification should be performed for the results where applicable and the number of cells imaged/quantified reported. The images are convincing, but quantification is still required to fully substantiate the claims (Figure 2E, Figure 3, Figure 4B).

2. Figure 7 and associated text: The authors make statements regarding the kinetics of the reaction; however, they do not quantify these results. Please quantify the kinetics to support these claims.

3. Figure 7: To further delineate between the USP10 and USP36 in processing of FUBI-eS30, USP10 can be included in in-vitro processing assay. The authors spoke towards the drawbacks of this assay in yielding false positive results due to the reaction occur outside of a cellular context. However, if USP10 is unable to process FUBI-eS30 in-vitro then it would enhance the claim the USP36 is a main enzyme for FUBI-eS30 processing.

4. To support USP36 as being the main protease for FUBI-eS30, the authors cite Rye et al. 2021. Although that reference shows USP36's involvement in pre-rRNA processing and translation, it is unclear if their results yielded a similar pre-40S defect as in this manuscript. To support USP36's role in FUBI-eS30 cleavage, ribosome biogenesis assays can be performed upon USP36 depletion. For example, polysome profiling, pre-rRNA processing Northern blots, and/or immunofluorescence (as in Figure 3) of RBFs can be performed to observe if USP36 depletion leads to similar defects as expression of the uncleavable FUBI-eS30 constructs. If it does not, then this strongly indicates that are other proteases or regulatory factors involved in this cleavage that remain to be studied.

*Reviewer #3 (Recommendations for the authors):*

I have no further recommendations and feel this study can be published immediately without revisions. It is well within the scope of *eLife* with respect to the importance of the problem, degree of advance, and quality of the research. The study was an enjoyable read.

---

## [Author Response]

Essential revisions:1) Over-expression of the uncleavable form of FUBI-eS30 results in several dominant negative phenotypes. This transgene could be acting as a sink for a variety of factors needed for ribosome biogenesis, in either a specific or non-specific manner. At the very least, this caveat should be mentioned/discussed.

Our combined proteomics and cell biological analyses demonstrates that presence of the non-cleavable FUBI-eS30 mutants causes very specific and persistent late phenotypes in 40S subunit biogenesis, accompanied by a failure in the release of a set of a very specific subset of RBFs from pre-ribosomal particles in the cytoplasm (Figure 4). Based on the presented experiments, we see no reason to assume that free non-cleavable FUBI-eS30, i.e. the fraction of mutant FUBI-eS30 that is not part of pre-40S particles, could serve as a ‘sink’ for late-acting RBFs, thereby indirectly causing the observed defects in 40S subunit maturation. For instance, in support of our conclusions, biochemical analysis of the sedimentation behaviour of the endonuclease NOB1, a late-acting RBF affected by the FUBI-eS30 mutants, demonstrates that this enzyme is further enriched on pre-40S particles upon expression of the non-cleavable FUBI-eS30 mutants, rather than titrated away into the top fractions of the gradient where unincorporated FUBI-eS30 mutants migrate (Figure 4—figure supplement 2C). We have now added a quantification of NOB1’s sedimentation behaviour as part of Figure 4—figure supplement 2. Similarly, although not systematically tested, we have not observed changes in the sedimentation behaviour of RIOK2, RIOK1 or PNO1 to the upper gradient fractions upon expression of non-cleavable FUBI-eS30. Importantly, we also note that in the chase experiments presented in Figure 4—figure supplement 2, there isn’t any visible free pool of mutant FUBI-eS30 after the chase period (Panel C); yet the cytoplasmic 40S subunit biogenesis defect remains very penetrant (Panel B). Furthermore, early maturation steps, which involve the vast majority of RBFs, remain by-and-large unaffected, as is 60S biogenesis, also arguing against pleiotropic unspecific effects of the expressed constructs. Thus, in light of the observed RBF recycling defects, biochemical data, and structural considerations regarding the expected position of uncleaved FUBI-eS30 on the 40S subunit, our interpretation that non-cleavable FUBI-eS30 mutants induce the accumulation of pre-40S particles that act as a specific ‘sink’ for the associated RBFs seems consistent with the data.

Still, to address the concern of the reviewer and as we have not formally excluded such scenario for all late-acting RBFs, we have added a sentence to the respective section on p. 11, to read:

‘Although we consider it unlikely, we cannot formally exclude that unincorporated, uncleaved FUBIeS30 could serve as an unspecific ‘sink’ for one or some of the affected RBFs. However, at least NOB1 does not change its sedimentation behaviour towards the pool of free protein in sucrose gradient analysis (Figure 4—figure supplement 2C and D), consistent with our conclusion.’

2) In Figures 3 and 4, imaging data is used to infer differences in ribosome biogenesis and/or association of certain factors with the pre-40S. Without biochemical experiments, the authors should temper their conclusions in the text regarding these data.

We note that some biochemical data had already been presented as parts of Figure 4—figure supplement 2 and Figure 5. Furthermore, we now support our conclusion of eS30 incorporation into nuclear pre-40S subunits by biochemical data (new panel B in Figure 3).

Moreover, we have more carefully phrased the respective conclusions, to read:

p.9: ‘Collectively, the nucle(ol)ar accumulation of FUBI(WT)-eS30-StHA upon induction of nuclear 40S maturation and export defects *suggests* that eS30 assembles into nuclear pre-40S ribosomes, likely already in the nucleolus.’

p.10: ‘In summary, the comprehensive examination of the (re)localization of various RBFs *suggests* that FUBI-eS30 cleavage is a prerequisite for recycling of the RBFs RIOK2, PNO1, NOB1, and RIOK1 in final steps of cytoplasmic 40S maturation.’

3) Figure 7 and associated text: The authors make statements regarding the kinetics of the reaction; however, they do not quantify these results. Please quantify the kinetics to support these claims.

In the revised version, we have included this quantification as new panel E in Figure 7.

4) To support USP36 as being the main protease for FUBI-eS30, the authors cite Rye et al. 2021. Although that reference shows USP36's involvement in pre-rRNA processing and translation, it is unclear if their results yielded a similar pre-40S defect as in this manuscript. While the involvement of USP36 in processing FUBI-eS30 is fairly clear, the authors should comment on the possibility that other DUBs may participate in this process.

We agree that this is a likely scenario and did in fact already mention this possibility in the discussion, to read:

‘Of course, it is also possible that there are other proteases working redundantly with USP36 on FUBI-eS30 processing.’ (Mind the change of ‘USPs’ to ‘proteases’ in the revised version).

‘A ULP confined to the nucleolus such as USP36 seems perfectly suited to prevent premature FUBIeS30 processing, although the existence of redundant FUBI-specific ULPs in other cellular compartments cannot be excluded at this point.’

5) Pg 19, paragraph 2: The authors' state "uncleaved FUBI-eS30 USP36 depletion was at least ten times lower compared to the expressed non-cleavable FUBI-eS30 constructs". Please reference the figure or present the data to support this statement as this is an important aspect of the experimental design for the reader to know.

To address the point of the reviewer, we have included a panel showing the quantification of the corresponding data in Figure 6 (new panel 6D).

In the respective HeLa cell line used for this quantification, where the expression from the transgenes is comparable to that of endogenous FUBI-eS30 (Figure 1C), the difference in levels of the non-cleavable FUBI-eS30 mutant and the uncleaved endogenous FUBI-eS30 upon USP36 depletion is even larger than a factor of ten. Since we are aware that the exact ratio between both forms largely depends on how efficient USP36 depletion is in different cellular backgrounds, we have rephrased the respective sentence in the discussion, to read:

‘However, the level of endogenous uncleaved FUBI-eS30 upon USP36 depletion *is much lower than* that of the expressed non-cleavable FUBI-eS30 constructs (Figure 6D), indicating that USP36 depletion did not result in sufficient levels of uncleaved FUBI-eS30 to trigger a dominant-negative effect.’

Reviewer #1 (Recommendations for the authors):1) Ribosome biogenesis assays should be performed to test whether disruption of USP36 results in similar defects as expression of the uncleavable FUBI-eS30 constructs.

As already mentioned in the discussion of our manuscript, we had performed such experiments, not observing effects on the steady state localization of select RBFs upon USP36 depletion. We explain this result by the insufficient accumulation of endogenous FUBI-eS30 upon deletion of USP36, either because depletion of the enzyme is too inefficient or/and because there are redundant DUBs, as stated in the discussion and now shown in the new Figure 6D.

2) Over-expression of the uncleavable form of FUBI-eS30 results in several dominant negative phenotypes. This transgene could be acting as a sink for a variety of factors needed for ribosome biogenesis. At the very least, this caveat should be mentioned and elaborated on in the discussion. Do these cells grow at different rates versus the wild-type controls and/or parental lines? If so, these changes could be further feeding back onto ribosome biogenesis. Does altering the level of transgene expression change any of the described phenotypes. These points are important in regards to some of the conclusions made in the paper (e.g. page 10-" In summary, the comprehensive examination of the (re)localization of various RBFs indicates that FUBI-eS30 cleavage is a prerequisite for recycling of the RBFs RIOK2, PNO1, NOB1, and RIOK1 in final steps of cytoplasmic 40S maturation."

Please refer to our response to Essential Points #1.

With regards to the growth of cells expressing non-cleavable FUBI-eS30 constructs compared to wildtype control or parental cells, we have not observed substantial differences in growth rates within the short durations of the experiments, i.e. expression of the constructs of up to only 24 hours. Based on the observed ribosome biogenesis defects, we think that it is reasonable to assume that prolonged expression of the non-cleavable constructs may negatively affect cell proliferation.

3) Figure 1C is a bit unconvincing. Are the endogenous protein levels unaltered by expression of the transgenes? Can the authors provide some quantification?

In Figure 1C, we analyzed the expression of the tagged FUBI-eS30 derivatives 17 hours after tetracycline induction. At this early time point, one would not yet expect a substantial effect on the levels of the endogenous proteins, as pre-existing subunits are stable, and we have thus not quantified the data. We demonstrate in other Figures (Figure 1D, Figure 4—figure supplement 2) that the expressed tagged versions are efficiently incorporated into 40S subunits, indicating that they efficiently compete with the endogenous protein.

4) Imaging data in Figure 3 is used to infer that eS30 joins pre-40S ribosome subunits in the nucleus. Pulldown of eS30 with nuclear pre-40S is needed to strengthen this conclusion.

To address this point, we have performed pull-down experiments from HEK293 cells using tagged C21orf70, a nucle(ol)ar RBF (Zemp et al., 2014), as a bait (new panel Figure 3B). Immunoblotting of the eluates demonstrates that eS30 is indeed associated with pre-40S particles isolated by this pulldown. In addition, we now also refer to previous evidence obtained by mass spectrometric analysis of the nucleolar proteome, to read:

‘This conclusion is also supported by previous mass spectrometric data that indicated an efficient accumulation of newly synthesized FUBI-eS30 in isolated nucleoli (Lam et al., 2007). […] Analysis of the pull-down eluates by immunoblotting revealed that eS30 is indeed associated with these nucle(ol)ar 40S precursors (Figure 3B) that also contained the early pre-40S RBF NOC4L, as expected (Zemp et al., 2014), and lacked RPS26, which is incorporated into pre-40S subunits in the cytoplasm (Ameismeier et al., 2020; Plassart et al., 2021).’

5) On Page 9, the authors state "Strikingly, PNO1, an RBF that localizes to nucleoli under steady state conditions, relocated to the cytoplasm. Similarly, a prominent recycling effect was also evident for RIOK2 and NOB1. Both are cytoplasmic at steady state but in cells expressing non-cleavable FUBI-eS30-StHA, they failed to accumulate in the nucleus upon LMB treatment, reflecting a failure in their timely release from cytoplasmic pre-40S subunits (Zemp et al., 2009)." Biochemical experiments are needed to confirm association/ dissociation with the ribosome.

Our previous biochemical experiments published in Zemp et al., JCB, 2009 and Zemp et al., JCS 2014 have established that the described microscopically evident changes in the steady state localization or dynamic shuttling behaviour of these RBFs reflect their retarded release from pre-40S particles, as demonstrated biochemically by sucrose gradient analyses of the sedimentation behaviour of these RBFs. We have now also added the second citation to our previous work. Furthermore, we note that we provide biochemical evidence for one of the analyzed factors (NOB1) in Figure 4—figure supplement 2, for which we have now added a quantification (Panel C).

6) Typo on page 47- brillinat should be brilliant

Thanks for spotting the mistake. Corrected.

Reviewer #2 (Recommendations for the authors):1. In situ hybridization and immunofluorescence images do not have a nuclear and/or nucleolar staining associated with them, but claims are made to the localization and abundance of the fluorescence within these cellular compartments. Please indicate with the appropriate nuclear and/or nucleolar staining markers. In the Materials and methods Section it was reported that Hoechst was used, please report those images for the nuclear marker. ENP1 seems to serve as a nucleolar marker since co-stained with anti-HA (Figure 3). Additionally, nuclear vs cytoplasmic signal quantification should be performed for the results where applicable and the number of cells imaged/quantified reported. The images are convincing, but quantification is still required to fully substantiate the claims (Figure 2E, Figure 3, Figure 4B).

To address the request of the reviewer, we have now included Hoechst staining as a nuclear marker in Figures 2 and 4.

ENP1 is indeed a nucleolar maker at steady state but accumulates in the nucleoplasm along with newly synthesized subunits upon treatment of cells with LMB, as explained in the text. In those panels, in which a nucleolar marker is needed for the interpretation of the results (i.e. si-eS4x), ENP1 localized to nucleoli, and hence an additional nucleolar marker seemed obsolete to us at the time when we conceived the experiment. Please note that nucleoli don’t disintegrate under the chosen experimental conditions for LMB treatment (Badertscher et al., Cell Reports, 2015), and that our conclusion on incorporation of eS30 into nuclear pre-40S subunits is now supported by biochemical data (new Panel 3B).

To address the request of the reviewer, we have now added quantifications to Figures 2 and 4 (new Panels 2F and 4C). We could not quantify nuclear vs cytoplasmic signal since for many conditions there is no appropriate cytoplasmic signal/marker for the detection of cellular contours, as all cells exclusively display a nuclear signal for the analyzed RBFs. Therefore, we have decided to rather bin cells into different categories based on the striking phenotypic differences for the quantitative analyses.

2. Figure 7 and associated text: The authors make statements regarding the kinetics of the reaction; however, they do not quantify these results. Please quantify the kinetics to support these claims.

We have added this quantification, supporting our claim.

3. Figure 7: To further delineate between the USP10 and USP36 in processing of FUBI-eS30, USP10 can be included in in-vitro processing assay. The authors spoke towards the drawbacks of this assay in yielding false positive results due to the reaction occur outside of a cellular context. However, if USP10 is unable to process FUBI-eS30 in-vitro then it would enhance the claim the USP36 is a main enzyme for FUBI-eS30 processing.

The depletion of USP10 did not cause any accumulation of uncleaved FUBI-eS30 in living cells, neither alone nor in combination with USP36. We were thus unsure whether eventually obtaining positive evidence for USP10’s activity in vitro would alone be sufficient to claim a role for this enzyme in FUBI-eS30 processing and we therefore did not invest efforts in recombinant expression and purification of USP10. For USP36, it is the combined evidence obtained in vivo and vitro which supports our claim that USP36 is a FUBI-eS30 processing enzyme.

4. To support USP36 as being the main protease for FUBI-eS30, the authors cite Rye et al. 2021. Although that reference shows USP36's involvement in pre-rRNA processing and translation, it is unclear if their results yielded a similar pre-40S defect as in this manuscript. To support USP36's role in FUBI-eS30 cleavage, ribosome biogenesis assays can be performed upon USP36 depletion. For example, polysome profiling, pre-rRNA processing Northern blots, and/or immunofluorescence (as in Figure 3) of RBFs can be performed to observe if USP36 depletion leads to similar defects as expression of the uncleavable FUBI-eS30 constructs. If it does not, then this strongly indicates that are other proteases or regulatory factors involved in this cleavage that remain to be studied.

We did not claim that USP36 is the only protease working on FUBI-eS30 and we had already suggested a potential redundancy with other ULPs in the discussion. Furthermore, we mention that we have already performed some of the suggested experiments and obtained negative data, as also mentioned in the discussion. As indicated in our response to Reviewer 1, we assume that accumulation of unprocessed endogenous FUBI-eS30 is insufficient to cause dominant-negative effects.